**communications** engineering

# Comparing strategies for the mitigation of SARS-CoV-2 airborne infection risk in tiered auditorium venues
S. Mareike Geisler [1,7] ✉, Kevin H. Lausch [2,7], Felix Hehnen[3,4], Isabell Schulz[3,4], Ulrich Kertzscher[3,4], Martin Kriegel [2], C. Oliver Paschereit[5], Sebastian Schimek[5], Ümit Hasirci[3], Gerrid Brockmann [2], Annette Moter[6], Karolin Senftleben[1] & Stefan Moritz [1] ✉

The COVID-19 pandemic demonstrated that reliable risk assessment of venues is still challenging and resulted in the indiscriminate closure of many venues worldwide. Therefore, this study used an experimental, numerical and analytical approach to investigate the airborne transmission risk potential of differently ventilated, sized and shaped venues. The data were used to assess the magnitude of effect of various mitigation measures and to develop recommendations. Here we show that, in general, positions in the near field of an emission source were at high risk, while the risk of infection from positions in the far field varied depending on the ventilation strategy. Occupancy, airflow rate, residence time, virus variants, activity level and face masks affected the individual and global infection risk in all venues. The global infection risk was lowest for the displacement ventilation case, making it the most effective ventilation strategy for keeping airborne transmission and the number of secondary cases low, compared to mixing or natural ventilation.

The severe acute respiratory syndrome coronavirus type 2 (SARS-CoV-2) is the causative agent of the coronavirus disease 2019 (COVID-19) and is transmitted primarily by infectious respiratory droplets and aerosols; alternative transmission pathways through direct contact or fomites can be considered of low epidemiological relevance[1,2]. One of the first and longest lasting containment measures during the COVID-19 pandemic was the closure of venues around the world[3]. Both large- and small-scale events were assumed to increase the risk of virus transmission and thus amplifying the burden of the pandemic. In fact, there are many reports of transmission events in confined and poorly ventilated indoor spaces, partly due to infectious aerosols[4,5]. However, recent studies have shown that the event-related risk of contracting SARS-CoV-2 can kept very low with well-functioning ventilation systems and appropriate mitigation strategies to reduce exposure to infectious aerosols[6–9].

Ventilation strategies in venues are very heterogeneous and include a variety of displacement (DV), mixing (MV) and natural (NV) ventilation concepts. The room-specific airflow and consequently the accumulation of

airborne pathogens is strongly influenced by the ventilation strategy and the different ways in which air is supplied and extracted[10,11]. In DV systems, air is supplied at low velocity above the floor directly to the occupied zone, rises due to buoyancy effects and is exhausted at the ceiling. MV systems introduce air at high velocity from the ceiling or side wall outside the occupied zone to mix with the indoor air and dilute contaminants, which are then exhausted. Unlike mechanically ventilated rooms, NV systems use only natural forces such as wind or buoyancy effects to create air movement and to supply fresh air. There are a lot of studies, which reported that DV systems are considered to have a lower risk of airborne disease transmission than MV or NV systems due to the higher ventilation effectiveness[10,12–16]. Other authors, however, have reported contradictory results[17,18], but highlighted the need for a sufficient ventilation rate of ≥3 air changes per hour (ACH) to effectively reduce the risk of infection with DV[18,19]. Venues are usually complex spaces with multiple areas that require special ventilation concepts to ensure good air quality and a low risk of infection throughout the venue. In the past, however, mechanical ventilation systems of venues were given a

[1]Section of Clinical Infectious Diseases, University Hospital Halle (Saale), Ernst-Grube Str. 40, 06120 Halle (Saale), Germany. [2]Institute of Energy Technology, Department Energy, Comfort and Health in Buildings, Technical University of Berlin, Marchstraße 4, 10587 Berlin, Germany. [3]Biofluid Mechanics Laboratory, Institute of Computer-assisted Cardiovascular Medicine, Deutsches Herzzentrum der Charité, Augustenburger Platz 1, 13353 Berlin, Germany. [4]Charité – Universitätsmedizin Berlin, corporate member of Freie Universität Berlin and Humboldt-Universität zu Berlin, Charitéplatz 1, 10117 Berlin, Germany. [5]Institute of Fluid Dynamics and Technical Acoustics, Hermann-Föttinger-Institute, Chair of Fluid Dynamics, Technical University of Berlin, Müller-Breslau-Str. 8, 10623 Berlin, Germany. [6]Charité – Universitätsmedizin Berlin, Institute of Microbiology, Infectious Diseases and Immunology, Hindenburgdamm 30, 12203 Berlin, Germany. [7]These authors contributed equally: S. Mareike Geisler, Kevin H. Lausch. ✉e-mail: mareike.geisler@uk-halle.de; stefan.moritz@uk-halle.de

low priority in the prevention of airborne diseases, as the focus was primarily on the requirements for quiet operation, thermal comfort and economical energy consumption[20–24]. Although venue studies on the risk of airborne disease transmission have increased since the COVID-19 pandemic, a comprehensive risk assessment comparing and classifying different ventilation concepts with regard to their risk of transmitting infectious aerosols is still lacking. The only large-scale monitoring study analyzed the ventilation effectiveness in up to 10 differently sized and ventilated theatres during 90 regular events with spectators using $CO_2$ sensors[25,26]. However, the lack of controlled study conditions, as well as the general inability of $CO_2$ approaches to account for the effectiveness of face masks, air purifiers and the infectivity of individuals, e.g. high emitters[27], indicated that further research is needed. Few studies examined SARS-CoV-2 transmission via aerosols using analytical[28,29], computational fluid dynamics (CFD)[6] or experimental models[25,26,30] for single venues. The analytical approach, such as the Wells-Riley or dose-response approach, assumes, that aerosols are instantaneously and uniformly distributed in space[31]. Consequently, the spatio-temporal distribution of aerosols is neglected, resulting in the same risk of infection for every person in the room, regardless of their position. CFD analysis can overcome this problem by simulating and visualizing venue-specific aerosol distribution patterns, thus enabling the calculation of individual infection risks, as recently done by several published CFD based studies[11,32–34]. Limitations of this approach are the simplified assumptions of stationary airflow patterns, boundary conditions and ideal airborne particles. Therefore, experimental measurements are needed for the validation of CFD data and vice versa. Current methodologies use optical systems, $CO_2$, tracer gas, artificial aerosols or virus surrogates to investigate infectious aerosol distribution in venues[25,30,35–38]. However, direct, fast and easy measurement of sputum-like aerosol particles in the immediate vicinity of the emission source and at various far-field positions in everyday environments is still challenging. The Aerosol Transmission Measurement System (ATMoS) fills the gap, as it can easily quantify aerosol and droplet transmission between dummies in real time and with high resolution at different environmental positions, even over large distances[39]. ATMoS enables room aerosol distribution and exposure measurements, making it suitable for the assessment of various indoor scenarios like different ventilation settings and mitigation strategies[40].

This large-scale approach has used a combination of experimental, numerical and analytical investigations to assess the airborne transmission risk potential of venues with different ventilation strategies. Therefore, ATMoS and CFD analyses were used in four different venues, one with displacement ventilation (DVV), one with mixing ventilation (MVV), one with natural ventilation (NVV) and one with a hybrid ventilation strategy combining displacement and natural ventilation (HVV). Numerical (CFD) and experimental (ATMoS) measurements were conducted for three different positions (front, middle, back) of the emission source. With CFD, the individual ($P_{CFD}$) and global ($R_{CFD}$) infection risks were calculated for each constellation. In addition, the global risk of infection derived from the experimental data ($R_{ATMoS}$) and the classical analytical Wells-Riley approach[30] ($R_{analyt}$) were assessed for each venue. Furthermore, different emission types, varying boundary conditions (e.g. occupancy, air flow rate) and variants of SARS-CoV-2 were taken into account for the risk analyses. The experimental measurement setup and the venue-specific data on aerosol amounts are presented in Schulz et al.[40], while the focus of this article is on the calculation of venue-specific infection risks. Consequently, the results were used to identify critical areas and conduct a ventilation-specific risk assessment, followed by a set of venue- and ventilation-specific recommendations to ensure safe events in future.

## Results
### Spatial distribution of individual infection risks in different ventilated venues using CFD analyses
To obtain aerosol distribution data for the entire location and for every position in the audience, CFD analyses were performed for four different ventilation scenarios in multi-tiered, seated indoor event locations (DVV, MVV, NVV and HVV). Using the concentration of infectious quanta of each occupants breathing zone, the venue-specific individual ($P_{CFD}$) and global risk of infection ($R_{CFD}$) could be calculated for different settings and emitter positions. Furthermore, an individual acceptable risk of infection $R_{acc}$ was determined for numerically derived infection risks and set at $10^{-2}$.

**Infection risk for a sedentary, passive emitter.** Figure 1 shows the infection risks for different positions in DVV, MVV and NVV. For DVV, a directional aerosol distribution with a pronounced aerosol plume

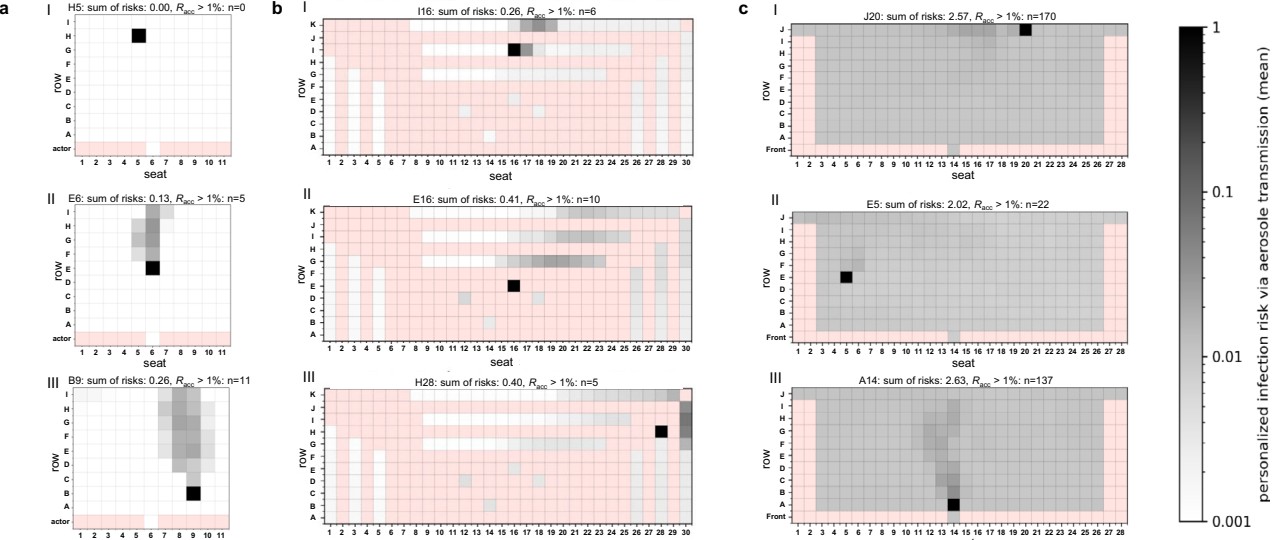

**Fig. 1 | Distribution of the numerically derived individual ($P_{CFD}$) and global risk of infection ($R_{CFD}$) for the venues with displacement (DVV), mixing (MVV) and natural (NVV) ventilation considering a sedentary, passive emitter.** Infection risk plots for the venues with displacement ventilation (**a**), mixing ventilation (**b**) and natural ventilation (**c**) are shown. **a** At DVV, the emitter was located at H5 (I), E6 (II) and B9 (III). **b** MVV emitter positions were at I16 (I), E16 (II) and H28 (III). **c** The positions of the NVV emitters were at J20 (I), E5 (II) and A14 (III). The individual risk of infection is plotted for each position, except for the red positions, as these do not represent seats in the audience. The sum of risks for each venue and emitter position as well as the number of spectators with the acceptable risk of infection $R_{acc} > 1\%$ is indicated above the plots. Figure 1aI-III, 1bI-III and 1cIII were adapted from a previously published manuscript in Schulz and Hehnen et al.[40].

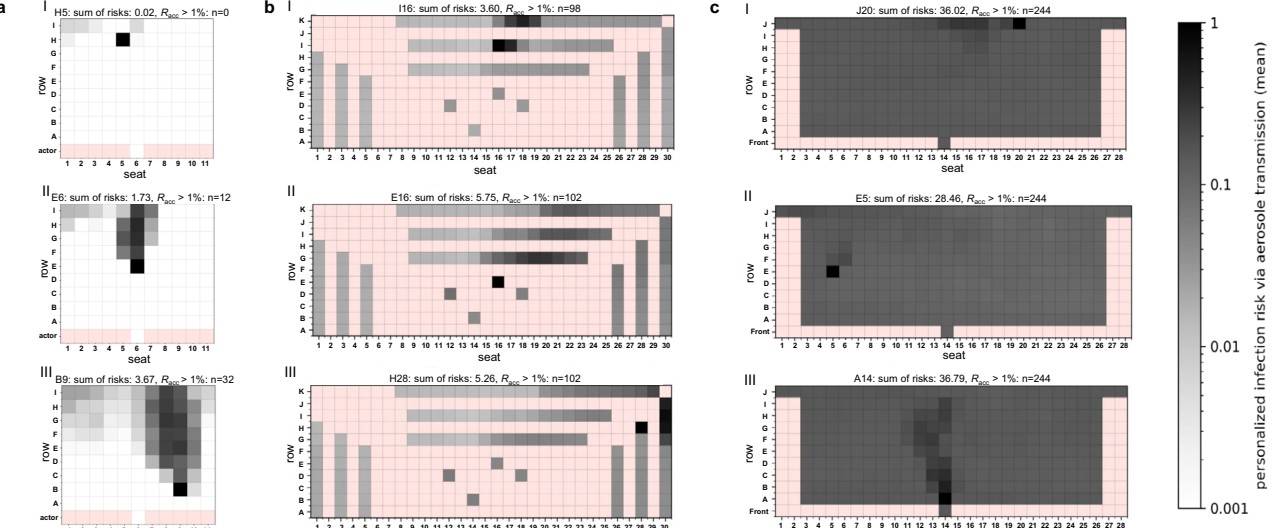

**Fig. 2 | Distribution of the numerically derived individual ($P_{CFD}$) and global risk of infection ($R_{CFD}$) for the venues with displacement (DVV), mixing (MVV) and natural (NVV) ventilation considering a high emitting individual (90th percentile).** Infection risk plots for the venue with displacement ventilation (**a**), mixing ventilation (**b**) and natural ventilation (**c**) are shown, considering a high emitter. **a** At DVV, the emitter was located at H5 (I), E6 (II) and B9 (III). **b** MVV emitter positions were at I16 (I), E16 (II) and H28 (III). **c** The positions of the NVV emitters were at J20 (I), E5 (II) and A14 (III). The individual risk of infection is plotted for each position, except for the red positions, as these do not represent seats in the audience. The sum of risks for each venue and emitter position as well as the number of spectators with the acceptable risk of infection $R_{acc} > 1\%$ are indicated above the plots.

behind the emitter was observed (Fig. 1aII+III, Fig. S3A). Thus, the seats behind the emitter were the most exposed, while the positions in front of and next to the emitter remained almost unaffected. Depending on the different emitter positions, the number of people exceeding the acceptable risk ($R_{acc}$) threshold of $10^{-2}$ ranged from 0 to 11 ($\triangleq$ 0–11% of the audience).

For MVV, a preferential but less directional flow of aerosol particles backwards towards the upper right corner was identified (Fig. 1bII-III, Fig. S3B). Positions with increased $P_{CFD}$ were located in all directions around the emitter and resulted in 5 to 10 spectators ($\triangleq$5–10% of the audience) reaching $R_{acc}$ above $10^{-2}$ in the near- and far-field of the emitter.

NVV with an ascending spectator area showed a directional aerosol distribution with a pronounced aerosol plume behind the emitter, especially for the emitter position A14 (Fig. 1cIII + Fig. S3D), like DVV. Seats behind the emitter had the highest $P_{CFD}$ values, but unlike DVV, the emission of aerosol particles resulted in contamination of the entire venue resulting in 22 to 170 spectators ($\triangleq$ 9–70% of the audience) exceeding $R_{acc}$.

The numerically derived global risk of airborne transmission $R_{CFD}$, corresponding to the number of new COVID-19 infections, also referred to as secondary cases, varied depending on the position of the emitter and the ventilation strategy. DVV showed the lowest $R_{CFD}$ values compared to MVV and NVV ranging from 0.00 to 0.26 (Fig. 1aI–III). For MVV the number of new COVID-19 cases was slightly higher, ranging from 0.26 to 0.41 (Fig. 1bI–III). $R_{CFD}$ for NVV was about 2 to 2.6 (Fig. 1cI–III).

**Infection risk for a high emitter (90th percentile).** In the case of a high emitter, the distribution of aerosol particles and $P_{CFD}$ was similar and dependent on the position of the emitter as for a passive emitter for all ventilation strategies studied (Fig. 2). The zone of increased risk was much more pronounced and wider for the DVV and NVV cases. In general, an enhanced release of infectious aerosol was associated with an increase in individual and global infection risk at all venues. For DVV, the number of spectators above $R_{acc}$ remained unchanged for H5 but increased by 2.4 to 2.9 times for E6 and B9, representing 12% to 32% of spectators (Fig. 2aI–III). At MVV and NVV, a high emitter resulted in the distribution of aerosol particles throughout the venue. This was

associated with increased $P_{CFD}$ values at all positions, as demonstrated by almost 100% of spectators achieving $R_{acc}$ above 1% (Fig. 2bI-III, 2cI–III).

In comparison with a passive emitter, $R_{CFD}$ increased by a factor of ~13 to 14 for all emitter positions at all venues (Fig. 2a–c). For DVV the number of secondary infections was highest for B9 with 3.67 and lowest for H5 with 0.02 (Fig. 2aI+III). $R_{CFD}$ ranged from 3.60 to 5.75 for MVV (Fig. 2bI-III) and 28.46 to 36.79 for NVV (Fig. 2cI-III).

**Experimental risk assessment (ATMoS)**
Table 1 summarizes the results of the global infection risk assessment based on ATMoS ($R_{ATMoS}$). Details of the experimental data are described elsewhere[40].

**Comparison of the experimental ($R_{ATMoS}$), numerical ($R_{CFD}$) and analytical ($R_{analyt}$) derived risk of infection for different ventilated venues**
The analytical risk of infection ($R_{analyt}$) was calculated for all venues and both emission modes according to Peng et al.[29] and was compared with $R_{ATMoS}$ and $R_{CFD}$ (Table 1). For DVV, both emission modes showed a good agreement for $R_{ATMoS}$ and $R_{CFD}$ in 3 out of 4 emitter positions. $R_{analyt}$ yielded the highest risk of infection compared to $R_{ATMoS}$ and $R_{CFD}$. In comparison, MVV showed a reasonable agreement for the values for $R_{ATMoS}$, $R_{CFD}$ and $R_{analyt}$. For the NVV position A14, the values for $R_{ATMoS}$, $R_{CFD}$ and $R_{analyt}$ were all markedly higher than for the other types of ventilation. However, while the NVV risk assessment based on analytical values revealed an approximate doubling of the values compared to MVV, the experimental and CFD values showed a ~5-fold or ~7-fold increase, respectively. Therefore, the agreement for NVV between the three approaches is worse than for DVV and MVV. A reason for this observation might be given by the steady-state correction factor $r_{ss}$ given in Peng et al.[29], which is used for $R_{analyt}$ but not for $R_{CFD}$.

**Risk evaluation of venues regarding different activity levels, variants of concern and mitigation strategies**
The CFD results were used to study the effects of different parameters on the number of COVID-19 secondary cases. The derived $R_{CFD}$ values were

compared to $R_{analyt}$ (Fig. 3). The reference case represented a 2 h event with full occupancy and full airflow considering the wild-type SARS-CoV-2 virus variant and no use of face coverings. Increased activity such as singing or shouting increased $R_{CFD}$ by a factor of 14.4 to 27.0 at all venues compared to a silent, passive emitter. The use of surgical masks reduced $R_{CFD}$ values by a factor of ~2 to 3 at all venues for both emission profiles. As a result of face mask usage, the number of new COVID-19 cases for a singing or shouting emitter decreased from up to 4.4 to 2.0 for DVV, 7.9 to 3.4 for MVV and 43.7 to 20.5 for NVV but were still about eight times higher than for a silent, passive, non-masked emitter. The use of FFP2/N95 masks reduced the $R_{CFD}$ values obtained with surgical masks by a factor of 7 to 10. Reducing the event duration to 1 h decreased the number of secondary infections by ~2 times, but still showed $R_{CFD} > 1$ for NVV. In contrast, increasing the residence time to 3 h resulted in 1.5-fold higher number of new COVID-19 cases. While $R_{CFD}$ for DVV and MVV remained <1, NVV showed $R_{CFD}$ values of 3 to 3.8. Note that the respiratory rate was set fixed for our study. However, halving

the respiratory rate and hence the exhalation volume flow would theoretically result in a halving of the emitted quanta which is comparable to a halving of the inhaled quanta dose of the occupants. This effect is indirectly and partly covered in Fig. 3 where the reduction of the duration to 1 h is listed.

Considering the variants of concern (VOC), the number of secondary infections increased 1.5, 2 or 3 times for Alpha, Delta or Omicron. In the case of DVV, $R_{CFD}$ remained below 1 for all three variants considered. For MVV, $R_{CFD} < 1$ was observed for the variants Alpha and Delta, while Omicron resulted in $R_{CFD}$ values of 1.04 to 1.25. Considering NVV, a silent and resting emitter infected with the Omicron variant resulted in 6.13 to 7.84 secondary infections.

The effect of reducing the airflow rate was investigated for DVV using CFD analysis, resulting in an increase of 1.7 to 9.86 in airborne infection risk.

### Special cases

**Hybrid venue (HVV).** HVV contains displacement-ventilated stalls and two naturally ventilated balconies. CFD analyses (Fig. 4, Fig. S3C) and experimental (Fig. S4) measurements observed a small aerosol plume with increased exposure behind the emitter for position R8S21. Spectators in front of and next to the emitter remained almost unaffected and showed low individual infection risks (Fig. 4, Fig. S4). A silent, sedentary emitter placed in the stalls spread infectious aerosols up to the balconies. On the contrary, aerosol emissions emanating from the balconies vanished slowly without exposing the stalls, but showed a 2.1 to 2.7 times higher risk of infection compared to the scenario where the emitter was placed in the stalls (Fig. 4a, Fig. S3C).

In the case of a high emitter, the aerosol partially dispersed over the entire balcony and led to a 13.6 to 14.3-fold increase in the airborne infection risk $R_{CFD}$ (Fig. 4b).

**Infectious actor.** In a special configuration of DVV, we placed the emitter on stage to experimentally simulate an infectious actor (Fig. 5aI). The aerosol that emanated from the infectious actor did not show a directional distribution with a pronounced aerosol plume, as seen for the emitter position B9 (Fig. 5aII). A silent, passive actor led to low individual infection risks and a low $R_{CFD}$ value of 0.07 (Fig. 5aI, Table 2). A high emitting actor resulted in the exposure of the entire venue, resulting in 1.02 secondary infections and 44 spectators exceeding the critical threshold $R_{acc}$ (Fig. 6aI, Table 2). In contrast, at emitter position B9, mainly spectators behind the emitter were exposed to infectious aerosols,

### Table 1 | Comparison of the experimentally ($R_{ATMoS}$), numerically ($R_{CFD}$) and analytically ($R_{analyt}$) derived global risk of infection

| Venue | Emitter position | Sedentary, passive | | | High emitter | | |
|---|---|---|---|---|---|---|---|
| | | $R_{ATMoS}$ | $R_{CFD}$ | $R_{analyt}$ | $R_{ATMoS}$ | $R_{CFD}$ | $R_{analyt}$ |
| DVV | B9 | 0.14 | 0.26 | 0.35 | 1.91 | 3.67 | 5.03 |
| | E2 | 0.14 | 0.12 | | 1.87 | 1.70 | |
| | E6 | 0.19 | 0.13 | | 2.59 | 1.73 | |
| | H5 | 0.04 | 0.00 | | 0.55 | 0.02 | |
| MVV | E16 | 0.33 | 0.41 | 0.35 | 4.57 | 5.75 | 4.88 |
| | H28 | 0.27 | 0.40 | | 3.76 | 5.26 | |
| | I16 | 0.23 | 0.26 | | 3.27 | 3.60 | |
| NVV | A14 | 1.48 | 2.63 | 0.79 | 20.01 | 36.79 | 11.27 |
| | E5 | – | 2.02 | | – | 28.46 | |
| | J20 | – | 2.57 | | – | 36.02 | |

The emitter positions investigated were B9, E6 and H5 in DVV, E16, H28 and I16 in MVV and A14, E5 and J20 in NVV for a silent, sedentary, and high emitter. To obtain $R_{ATMoS}$, the mean value of the seven absorber-specific $P_{ATMoS}$ values of one measurement was calculated and multiplied by the total number of spectators. $R_{CFD}$ represents the sum of the individual infection risks $P_{CFD}$. $R_{analyt}$ was calculated according to Peng et al.[29]. The global risk of infection is related to the number of COVID-19 secondary infections in the venue.

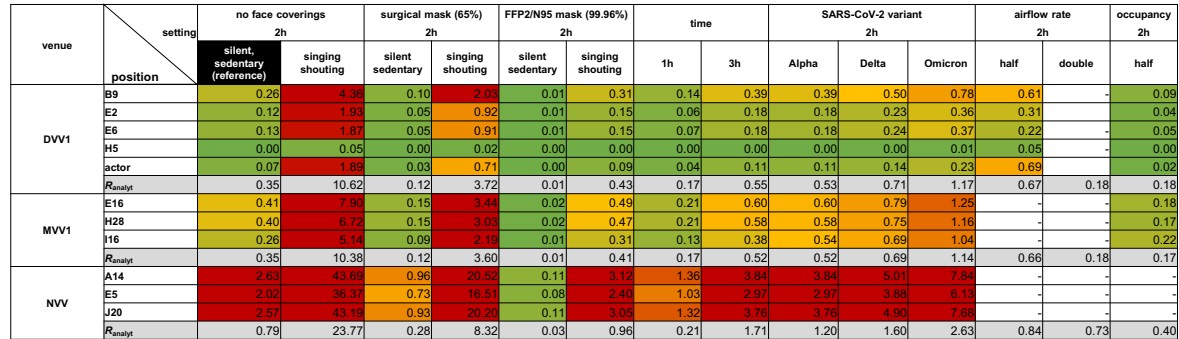
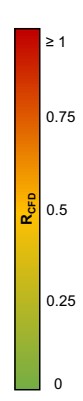

**Fig. 3 | Influence of different parameters and mitigation strategies on the global risk of infection $R_{CFD}$ of venues with different ventilation concepts.** The parameter study was performed for different parameters, mitigation strategies, activity levels and emitter positions for the venues with displacement (DVV), mixing (MVV) and natural (NVV) ventilation for the case of a single silent, sedentary emitter (reference). The global risk of infection $R_{CFD}$ represents the sum of the individual infection risks $P_{CFD}$ and was calculated for each emitter position of each setting. The $R_{CFD}$ value is a measure of the number of secondary infections. The reference settings (black) were as follows: no face masks, event duration of 2 h, SARS-CoV-2 wild-type variant, full airflow rate and full occupancy. The efficacy of masks was investigated using surgical masks (65% filtration efficiency (0.35)) and well-fitting FFP2/N95 masks (96% filtration efficiency (0.04)). $R_{CFD}$ values were highlighted according to their risk potential using a colour-coded scale with: high risk—$R_{CFD} \geq 1$ red, medium risk—$R_{CFD} = 0.5$–1 shades of yellow-orange-red, low risk—$R_{CFD} = 0$–0.49 shades of yellow-green. The analytical risk of infection $R_{analyt}$ values (grey) were calculated according to Peng et al.[29]. Empty boxes indicate the absence of numerical measurements for a given configuration.

**Table 2 | Sum of risks $R_{CFD}$ for the emitter stage position (actor), 50% reduced air flow rate and checkerboard pattern seating arrangement for different emitter positions at DVV**

| Emitter positions | Sedentary, passive | | | High emitter | | |
|---|---|---|---|---|---|---|
| | 100_100 | 100_chess | 50_100 | 100_100 | 100_chess | 50_100 |
| B9 | 0.26 | 0.09 | 0.61 | 3.67 | 1.32 | 8.36 |
| E2 | 0.12 | 0.04 | 0.31 | 1.70 | 0.54 | 3.99 |
| E6 | 0.13 | 0.05 | 0.22 | 1.73 | 0.71 | 3.11 |
| H5 | 0 | 0 | 0.05 | 0.02 | 0.01 | 0.64 |
| actor | 0.07 | 0.02 | 0.69 | 1.02 | 0.34 | 9.68 |

Supplementary table to Fig. 5 and Fig. 6 shows the sum of risks for the emitter positions B9, E2, E6, H5 and actor (on stage) for the silent passive emitter and the high emitter (90th percentile) with: standard conditions with full air flow rate (4500 m³/h) and occupancy rate (99 spectators) indicated as 100_100, checkerboard seating arrangement indicated as 100_chess and 50% reduced air flow rate indicated as 50_100.

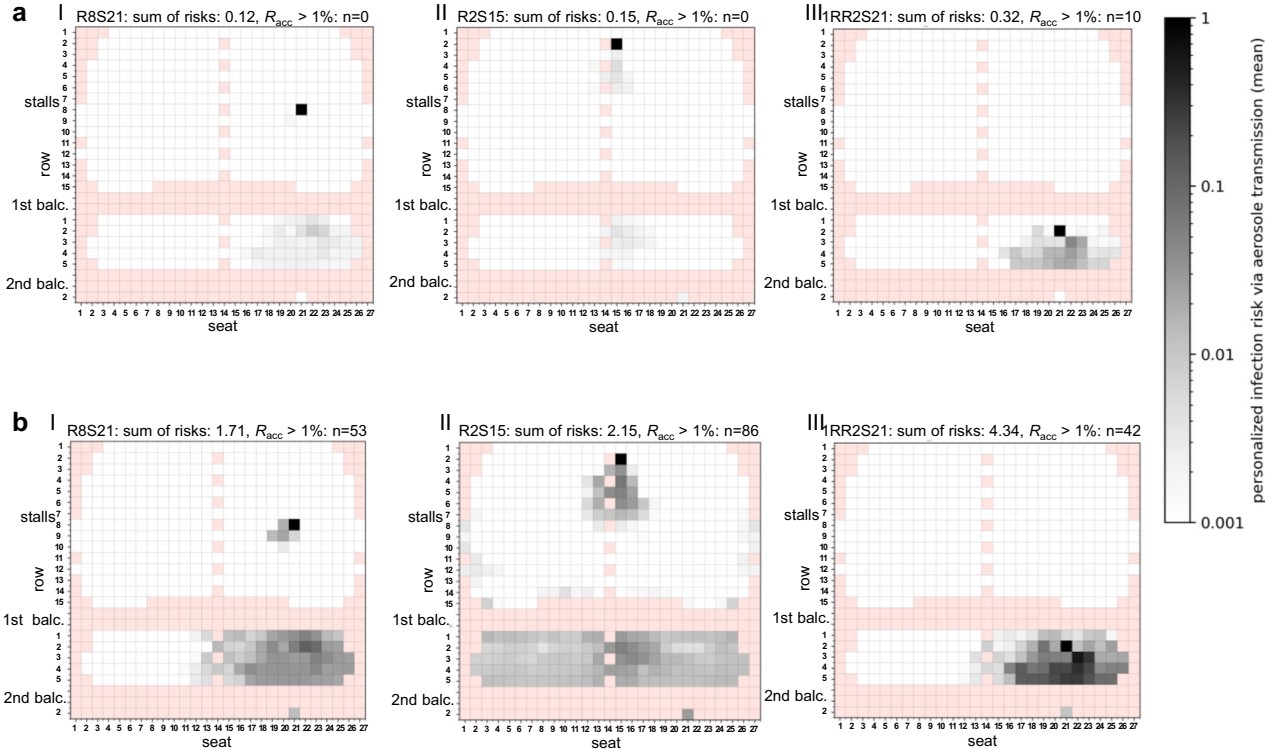

**Fig. 4 | Distribution of the numerically derived individual and global risk of infection for the venue with hybrid ventilation (HVV) with displacement ventilation in the stalls and unventilated balconies. a, b** Numerically derived infection risk plots for the emitter positions R8S21(I), R2S15 (II) and 1RR2S21 (III) for the silent passive emitter (**a**) and the high emitter case (90th percentile; **b**) are shown.

The individual risk of infection is plotted for each seat, except for the red shaded positions, as these do not represent seats in the audience. The sum of risks for each venue and emitter position as well as the number of spectators with the acceptable risk of infection $R_{acc} > 1\%$ are indicated above the plots. The positions of the stalls, 1st balcony (balc.) and 2nd balcony are indicated in the graphs.

resulting in 3.6-times more secondary infections than in the case of the infectious actor, but showed a lower number of people above $R_{acc}$ (Fig. 6aII, Table 2).

**Variations of boundary conditions**
**Reduced airflow rate.** CFD analyses were performed for DVV with a 50% reduced airflow rate (Figs. 5b, 6b). A pronounced aerosol plume was observed for emitter position B9, but infectious aerosols were also dispersed throughout the venue. This resulted in a doubling of spectators exceeding the $R_{acc}$ threshold of 1% and a 2.3-fold increase in secondary infections to 0.61 in the case of a silent, passive emitter (Fig. 5bII, Table 2). A silent, passive actor resulted in almost uniform exposure of the entire venue, increasing the global risk of infection by 9.9-fold to 0.69 (Fig. 5bI, Table 2). A high emitting spectator (B9) or actor, combined with a 50%

reduction in airflow rate, resulted in almost 100% of spectators exceeding $R_{acc}$ and 8.36 or 9.68 new COVID-19 cases (≙8–10% of the audience), respectively (Fig. 6bI–II, Table 2). This high-risk setting increased the global risk of infection by 32 (B9) and 138 (actor) times, respectively, compared with the low-risk setting with a silent, passive emitter in a well-ventilated venue. Thus, the formerly lower risk case of the actor exceeds the risk of the spectator case (B9) under circumstances of high emission profile and reduced airflow rate.

**Checkerboard pattern seating.** The effect of a 50% reduction in occupancy with checkerboard seating was investigated in venues with displacement (DVV) and mixing ventilation (MVV, MVV2) using ATMoS and CFD analyses (Figs. 5c, 6c, 3, Fig. S5–S8, Table 2). For DVV, the checkerboard pattern seating caused a broadening of the zone of

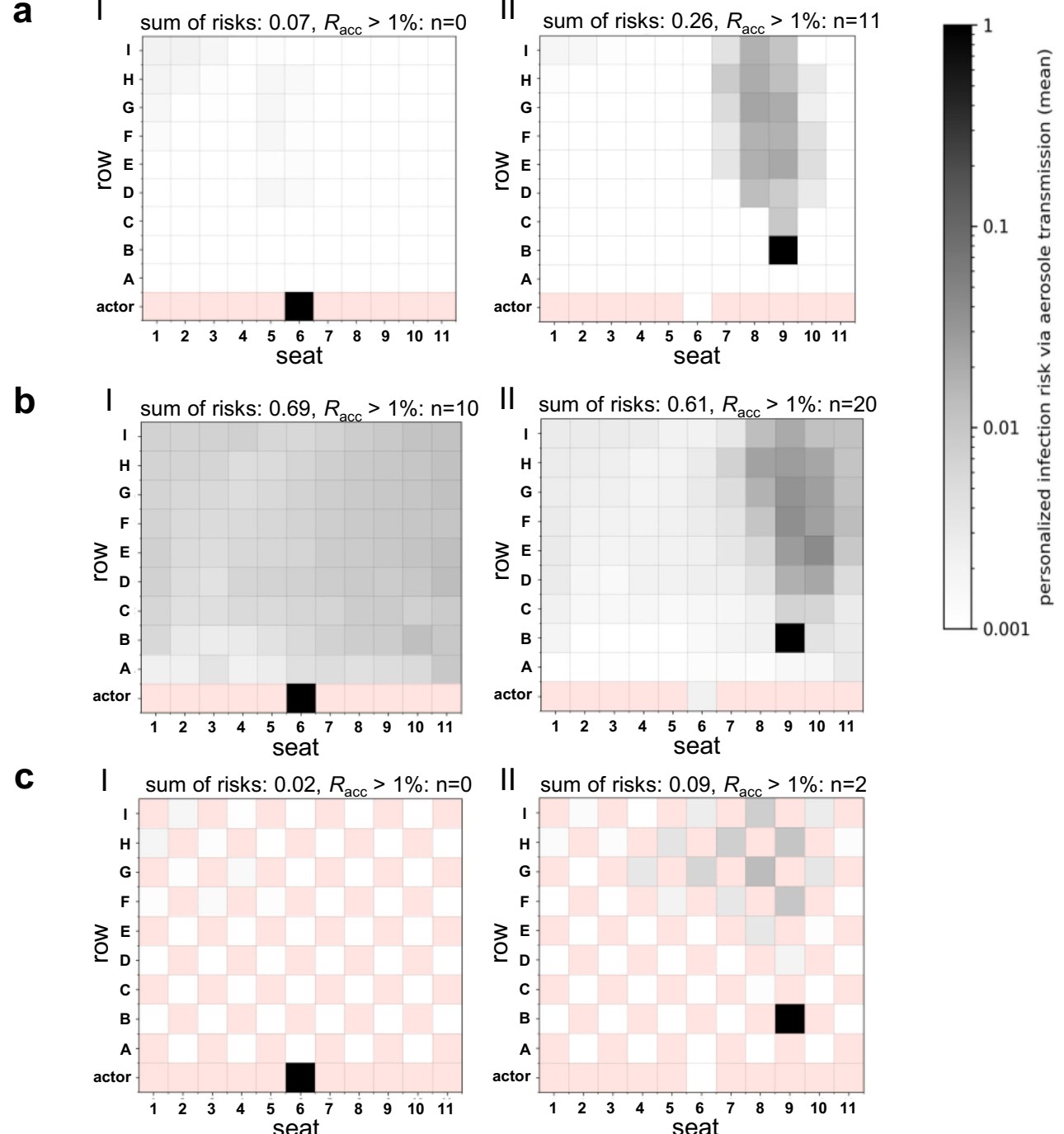

**Fig. 5 | Influence of the emitter stage position (actor), 50% reduced airflow rate and checkerboard pattern seating arrangement on individual ($P_{CFD}$) and global risk of infection ($R_{CFD}$) at the venue with displacement ventilation (DVV) for a silent passive emitter.** Infection risk plots for the stage (actor) (I) compared to the audience emitter position B9 (II) for the silent passive emitter are shown: (**a**) standard conditions with full air flow rate (4500 m³/h) and occupancy rate (99 spectators), (**b**) 50% reduced air flow rate and (**c**) checkerboard seating arrangement. The individual risk of infection is plotted for each position, except for the red positions, as these do not represent seats in the audience. The sum of risks for each venue and emitter position as well as the number of spectators with the acceptable risk of infection $R_{acc} > 1\%$ are indicated above the plots.

elevated risk for emitter position B9, but otherwise showed a qualitatively similar distribution of individual infection risks $P_{CFD}$ for all emitter positions and emission profiles compared with the full seating arrangement (Fig. 5cI–II, Fig. S5). $R_{CFD}$ values of emitter positions investigated in DVV were reduced by a factor of 2.6 to 3.5 with checkerboard seating (Figs. 5c, 6c, Table 2). For MVV, halving the occupancy rate showed numerically and experimentally a similar distribution of individual

infection risks $P_{CFD}$ and $P_{ATMoS}$ (Figs. S6, S7), but resulted in a different propagation risk pattern for MVV2 compared to full occupancy (Fig. S8). While for MVV emitter position H28 and E16 the $R_{CFD}$ values more than halved, the $R_{CFD}$ value hardly reduced for emitter I16 (0.26 to 0.22) with the checkerboard seating arrangement (Fig. S6). In total, however, occupancy reduction is associated with a reduction in $R_{CFD}$ for DVV and MVV.

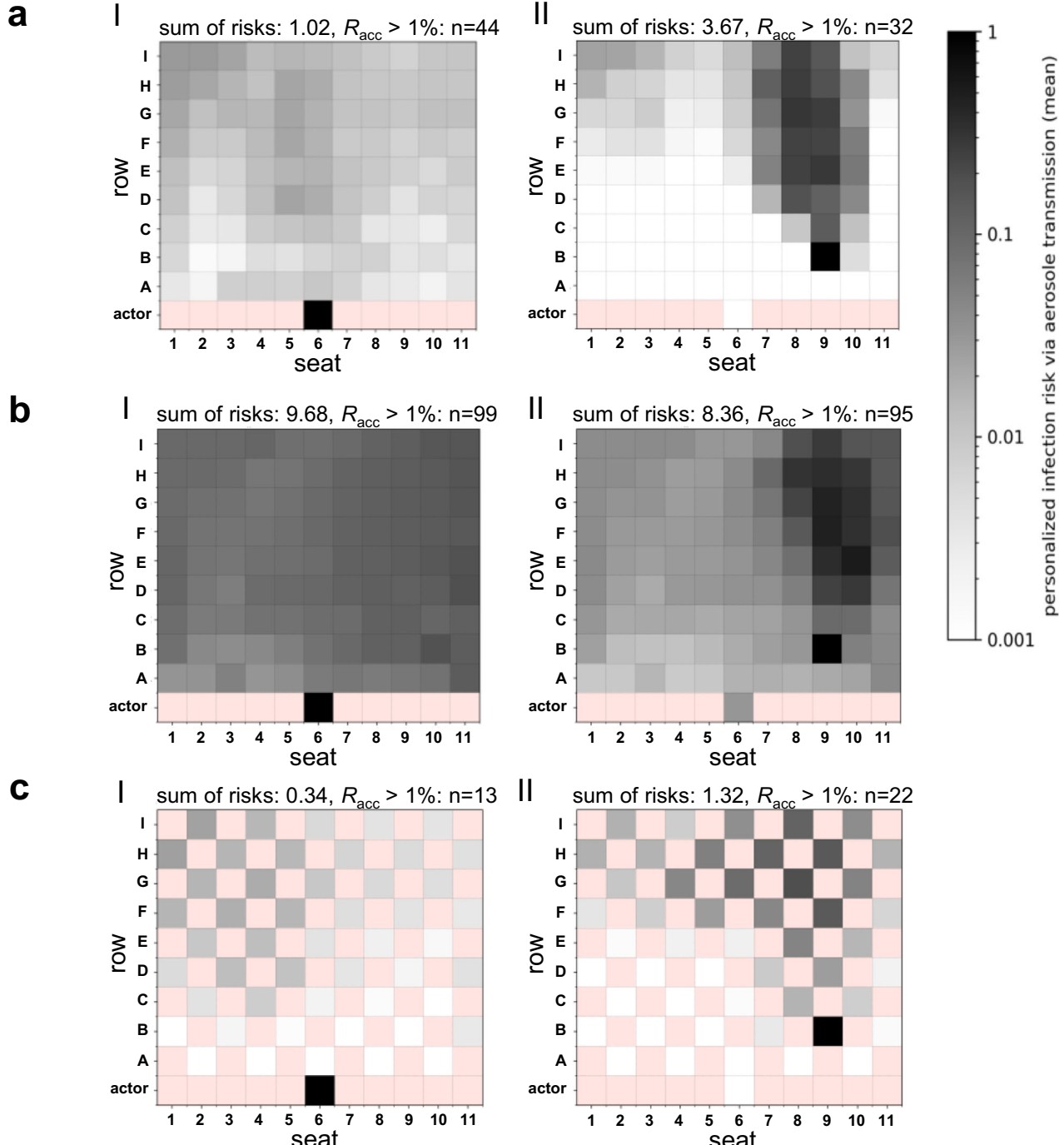

**Fig. 6 | Influence of the emitter stage position (actor), 50% reduced airflow rate and checkerboard pattern seating arrangement on individual ($P_{CFD}$) and global risk of infection ($R_{CFD}$) at DVV for a high emitter.** Infection risk plots for the stage (actor) (I) compared to the audience emitter position B9 (II) for the high emitter (90th percentile) are shown: (**a**) standard conditions with full air flow rate (4500 m³/ h) and occupancy rate (99 spectators), (**b**) 50% reduced air flow rate and (**c**) checkerboard seating arrangement. The individual risk of infection is plotted for each position, except for the red positions as these do not represent seats in the audience. The sum of risks for each venue and emitter position as well as the number of spectators with $R_{acc} > 1\%$ are indicated above the plots.

## Discussion

The COVID-19 pandemic has demonstrated that an appropriate risk assessment is needed to avoid the general and undifferentiated closure of venues in the future. To address this shortcoming, venues with different ventilation strategies were studied experimentally and numerically in terms of aerosol distribution and exposure to calculate venue-specific infection probabilities and risks compared to the classical analytical approach.

In our study, multi-tiered venues with displacement ventilation and an ascending spectator area posed a low risk of infection under a passive emitter scenario with 18.6 quanta h$^{-1}$, indicated by low individual transmission risks for most spectators and $R$ values well below one. However, the observed pronounced aerosol plume was associated with highly exposed positions in the near- and far-field behind the emitter, while the positions in front of and next to the emitter were almost unaffected, creating characteristic low- and high-exposure areas, similar to the results of previous studies[41–43]. The

expansion of the aerosol plume and subsequently the airborne transmission risk is strongly dependent on the position of an infectious person. Thus, our results confirm that this dependency – already shown for other (smaller) situations—is also true for large venues with displacement ventilation[15,42,44]. With mixing ventilation (MV), the airborne transmission risk was higher compared to DV. However, it was less dependent on the position of an infectious individual, as shown recently[34,45]. This indicates that—in contrast to DV—infectious aerosols were dispersed throughout MV venues, creating many low- and medium-risk positions and a few high-risk positions in the near- and far-field of the emitter. These findings were supported by Makris, Lichtner and Kriegel[41], who showed that the probability of inhaling aerosol particles at a distance of 1.5 m is twice as high and at a distance of 4 m four times as high for MV cases as for DV cases. Further MV studies have found high infection probabilities even at longer distances[11,46]. However, predicting highly exposed positions is more difficult as the airflow characteristics in MVV are less directional and likely to be sensitive to boundary conditions, as shown by the heterogeneous effects of the checkerboard seating arrangement at MVV2. Moreover, the influence of seasons[47], air temperatures[48] and spectator layout[49] on airflow characteristics has been demonstrated in recent studies on MV, but also on DV cases. Although the results of this study appear favourable towards DV systems, and despite being often applied in practice[50], a general recommendation of DV is not expedient. Bjørn and Nielsen[51] discuss the influential factors on contaminant distribution as e.g., motion, temperature gradient and mouth or nose exhalation. Similarly, Yuan et al. [52] point out, among other factors, limitations of DV with regards to the cooling load, space height, the wall (temperature) characteristic and the thermal comfort due to the vertical air temperature gradient. Additionally, Riffat et al. [53] emphasize the importance of an appropriate system design to avoid mixing polluted plumes and air in the occupied zone.

In the naturally ventilated venue NVV, the aerosol is distributed at high concentrations throughout the venue, regardless of the position of the infectious source, resulting in the highest airborne transmission risk for each emitter position compared to DV and MV venues, as shown here and previously[42]. Similarly, recent studies have shown that the risk of airborne transmission does not necessarily decrease with distance in naturally ventilated rooms, as the highest probabilities of infection were observed at longer distances, well beyond physical distance guidelines[11,46]. To keep $R < 1$ at NVV, the acceptable individual infection risk $R_{acc}$ must be reduced to 0.4% or the number of spectators to 92 (38%) or the exposure time to a maximum of ~43 min. The maximum number of spectators for a 1.5 h event is 121 (50%) (Table S3). Nevertheless, it should be noted that well-designed NV systems are potentially suitable for infection control and provide a cost-effective ventilation approach. However, they are usually highly dependent on natural forces such as wind, open windows or doors and air temperature, and are therefore characterized by unstable and changing airflow patterns, associated with a variety of potential distributions of infection risk in a room[10,54–57].

All high-emission scenarios, such as the more infectious SARS-CoV-2 variants, high viral loads or increased activity, were associated with an increased airborne transmission risk at all venues. For DVV, this was due to a much more pronounced and wider zone of elevated risks compared to a low-emission scenario. In contrast, a high emitter in MVV distributed the infectious aerosols throughout the venue. This resulted in a high risk of infection at any position in the venue, with almost all spectators exceeding the individual acceptable risk of infection, which is comparable to the high emitter scenario in the poorly ventilated DVV. A high-emitting spectator made NVV a high-risk site with a high potential for super-spreading events, regardless of the position of the infectious person. The maximum residence time or crowding index was markedly reduced to 3 min or six spectators to keep $R < 1$ (Table S3). The high emitter case demonstrated greater resilience for DVV compared to MVV or NVV. This was confirmed in a previous study, where even in high emission scenarios, DV outperformed MV systems, which spread the contaminant source over a larger part of the room[58]. In fact, high emitting individuals occur only occasionally, but given their

potential to provoke super-spreading events, they should be emphasized in risk assessment[59–62]. Furthermore, the cases of singing and shouting spectators are also high emitters and play a major role in Pop/Rock concerts and sporting events.

FFP2/N95 masks reduced the number of secondary cases by up to 26 times, turning DVV and MVV into minimal-risk sites and reducing the airborne transmission risk of NVV below the critical threshold for pandemic control of one. The risk-reducing effect of face masks on the transmission of COVID-19 has been shown by several studies, but its effectiveness is roughly limited by face-fitting and adherence[63–66]. In addition, their limits of effectiveness became apparent when considering VOC, virus-rich environments (e.g. hospitals) and prolonged residence in poorly ventilated areas (like NVV)[29,42,65,67]. This highlights the need for a combination of preventive measures. Since the risk of infection increases substantially with the duration of the event[46], it is recommended to limit the residence time in epidemic settings to a necessary minimum or to consider breaks[68]. Reducing the number of spectators is also an effective mitigation measure[24,69] as demonstrated by the results of the study. For DVV, the checkerboard seating arrangement resulted in a ~ 3-fold reduction in the global risk of infection, possibly indicating an additional downsizing effect beyond the effect of reduced audience size as shown recently[15]. However, at the moment, the cause of this downsizing effect is not definitely discernible.

A reduction in the airflow rate at DVV was associated with an up to ~10-fold increase in the risk of airborne transmission, reaching almost 100% of spectators above $R_{acc}$, which is in line with our previous results[6]. This shows that the effectiveness of DV is dependent on a room-appropriate and well-adjusted mode of operation, as previously demonstrated[18]. HVV underlines this, where a passive emitter in the mechanically ventilated stalls resulted in exposure to spectators in the distant unventilated balconies, perhaps small, but distinctly increased when a high emission scenario was considered. This highlights the need for good ventilation in all areas of the room. Adzic et al. [25] used $CO_2$ as a proxy for respiratory aerosols[27] and found higher levels in non-ventilated, but also in ventilated balconies. During the COVID-19 pandemic many venues were operated with 100% outdoor air and with maximum ventilation rates, resulting in high energy consumption and operating costs[25]. However, increasing flow rates do not necessarily reduce the risk of infection and close proximity exposure is still likely[44,70–72]. Therefore, ventilation modes and rates need to be optimized to balance the transmission risk and the operating costs.

In Fig. 7 we have summarized the available results of our study for ventilation type-specific recommendations for 1, 2 and 3 h events, targeting $R$ well below one. In the displacement ventilation case DVV, the use of surgical masks or a reduction in occupancy is recommended for events lasting longer than 3 h, when Omicron is considered. For MVV, recommendations for the use of face coverings are given for events of increasing duration, regardless of the infectivity of the virus variant. As surgical face masks were sufficient for the wild-type variant, FFP2/N95 masks were proposed for Omicron for 3 h events. The difficulty in predicting the risk of airborne transmission in naturally ventilated venues and the observed high risk of infection justify a general recommendation for FFP2/N95 masks. However, this may not be sufficient for prolonged exposure, particularly to virus variants with increased infectivity. Therefore, temporary closure of NV venues should be considered.

The special case of an infectious actor showed that a background actor (silent, passive) poses only a low risk of airborne transmission, whereas the risk increases dramatically in the more realistic scenario of a singing and shouting actor. In the case of a reduced airflow rate, a high-emitting actor can end up in a super-spreading event. The risk of infection from actors or singers should be more focused in the future to minimize the risk to the audience, but also to the ensemble, especially as in our experience the stage is often not connected to the ventilation system.

For the mixing ventilation case, the comparison of the three approaches revealed a good prediction of the overall airborne infection risk by the analytical approach, as shown recently[73,74]. For venues with DV or NV, however, the modified analytical Wells-Riley approach over- or

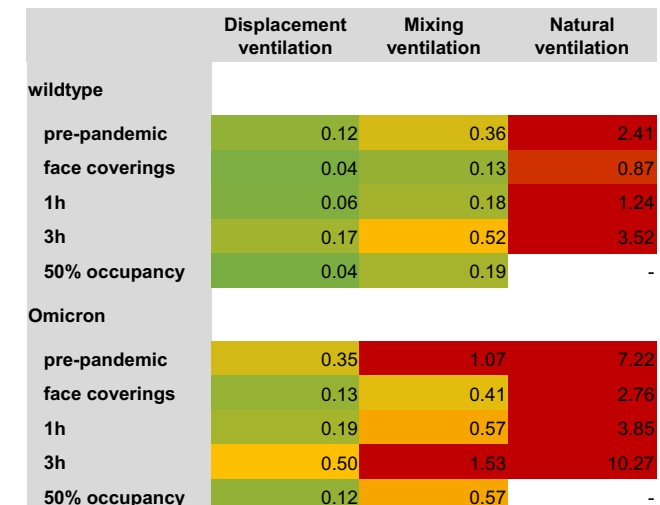
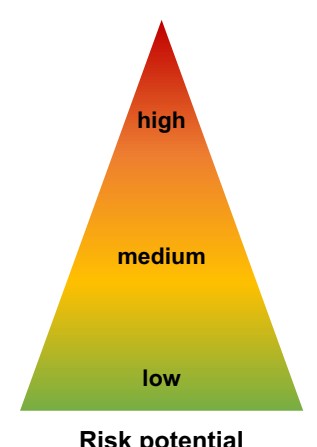

| a | Displacement ventilation | Mixing ventilation | Natural ventilation |
|---|---|---|---|
| **wildtype** | | | |
| pre-pandemic | 0.12 | 0.36 | 2.41 |
| face coverings | 0.04 | 0.13 | 0.87 |
| 1h | 0.06 | 0.18 | 1.24 |
| 3h | 0.17 | 0.52 | 3.52 |
| 50% occupancy | 0.04 | 0.19 | - |
| **Omicron** | | | |
| pre-pandemic | 0.35 | 1.07 | 7.22 |
| face coverings | 0.13 | 0.41 | 2.76 |
| 1h | 0.19 | 0.57 | 3.85 |
| 3h | 0.50 | 1.53 | 10.27 |
| 50% occupancy | 0.12 | 0.57 | - |

**Risk potential** (high / medium / low)

| b | wild-type | | | Omicron | | |
|---|---|---|---|---|---|---|
| | 1h | 2h | 3h | 1h | 2h | 3h |
| **Displacement ventilation** | - | - | - | - | - | surgical face mask reduction in occupancy rate |
| **Mixing ventilation** | - | - | surgical face mask | surgical face mask | surgical face mask | FFP2/N95 mask |
| **Natural ventilation** | surgical face mask | FFP2/N95 mask | FFP2/N95 mask | FFP2/N95 mask | FFP2/N95 mask | FFP2/N95 mask |

**Fig. 7 | Risk potential and recommendations of risk reduction strategies for venues with different ventilation concepts. a** The mean values of the venue-specific global risk of infection $R_{CFD}$ values of different emitter positions were calculated for various parameters and mitigation strategies considering the SARS-CoV-2 wild-type and Omicron variant. To keep the threshold for epidemic control $R < 1$, $R_{CFD}$ values were coloured according to their risk potential: $R \geq 1$ red, $R = 0.5$ yellow, $R = 0$ green. The values between the thresholds are coloured in shades of yellow-orange-red and yellow-green. The pre-pandemic settings with a silent, passive emitter were as follows: no face mask, event duration of 2 h and full occupancy. **b** Recommendations were given for the conduct of safe events, ensuring $R < 0.5$ for one-to-three-hour events, considering the SARS-CoV-2 wild-type and the Omicron variant for three ventilation concepts. For example, during 3 h events in a venue with displacement ventilation and Omicron variant $R < 0.5$ was achieved by using surgical face masks.

underestimated the airborne transmission risk, possibly indicating that the strong spatio-temporal dependence of the infection risk, caused by the directional flow patterns resulting from the ventilation, could not be captured by this approach. The differences are probably due to the random positioning of seven absorbers in the experimental approach compared to the consideration of the entire audience in the CFD approach. The absorber-specific aerosol concentrations vary strongly depending on their position to the emitter, making it difficult to select a representative set of absorber positions covering the full range of low and high-risk sites. Thus, the calculated venue-representative mean of experimental airborne transmission risks could be also biased because of the intermittent conditions experienced during the measurements/events which were not at all perfect laboratory conditions.

There are some limitations of the study. The differences between $R_{CFD}$ and $R_{analyt}$ observed for NVV are most likely particularly high because of the correction factor $r_{ss}$ as proposed by Peng et al.[29] for cases, where steady-state concentration is not reached. This is clearly a drawback of the simulation approach. In addition, ATMoS covered only a few positions at the venues studied, therefore a bias in the experimental results based on the selected positions cannot be excluded. The validity and reliability of the experimental data could be improved by using more absorbers and by repeated measurements. Furthermore, an event-related $R$-value threshold of 1 is too high, as an infectious person has additional contacts during the infectious period that must be included to estimate epidemic growth. Concerning the CFD simulations, the assumption of a steady state within the venues is arguable, although the duration of the events is on the order of a few hours. It is likely that during the event the local concentration would increase and converge to the steady state value, which is implied for the risk assessment during the complete event. To address this problem a correction factor $r_{ss}$ as proposed by Peng et al.[29] is considered for cases, where steady state concentration is not reached. This shortcoming could be resolved by the unsteady integration of the time-dependent, experienced doses on a stationary flow field or with a fully unsteady simulation approach. Due to the large time frame and the comparatively high temporal resolution, the latter approach would involve rather prohibitively high numerical resources for the given larger venues. Concerning the former approach, numerical experiments were conducted for particular cases, of which the largest part showed that values close to the steady state concentration were reached within a few minutes for the high-risk regions. As an exception, NVV showed slower increases in concentration due to its low air change rate, such that a correction factor for unsteady events, especially in background regions, could improve these particular approximations. Nevertheless, an additional benefit could not be expected from time-dependent integration results. Additionally, the assumption of constant thermal boundary conditions in a densely occupied event location is critical. A complete knowledge of all environmental boundary conditions and heat load reservoirs would certainly improve the accuracy of the numerical solution. Possibly, the effect would even outweigh the assumption of steady-state concentrations. In a similar way, event-specific, relevant boundary conditions (e.g., half-opened doors, reduced ventilation, intensified lighting) had to be disregarded, since they partly depend on the personal decision of the responsible technical staff or on the spectator's behaviour.

From a modelling perspective, the lack of evaporation and deposition effects in the CFD model is a limitation regarding general aerosol droplets. As a simplification, we imply for our aerosol to be perfectly airborne (in experiments and simulations), which may be partly justified by the aerosol size distribution measured in Schulz et al.[40] Arguably, the consideration of aerosol deposition could perhaps close the gap between the experimentally measured and simulated concentrations to some extent.

Furthermore, the risk models clearly depend on the precise estimation of shed quanta doses and their probability distribution. Due to the lack of a single log-normal distribution which fulfils all given requirements, tests have been conducted where μ was kept constant and σ varied to fit the lower or upper bounds. The effect on the resulting risk was negligible, especially, when compared to the risk differences imposed by a high emitter. However, it is crucial to note that the quanta emission rate applied in this study is itself based on certain assumptions as discussed for example in Peng et al.[29] and Buonanno et al.[75]

Since the respiratory viral load, which is relevant for quanta emission rate, is extremely heterogeneous across individuals and spans many orders of magnitude[61], the quality of risk prediction depends largely on the correct estimation of this influential factor. While this study used the best fit emission rate of Peng et al.[29] (18.6 quanta h$^{-1}$) and standard deviation as proposed by Buonanno et al.[75] to provide a combined log-normal distribution, other values would have been also justifiable.

Aganovic et al.[76] report the quanta emission rate to be possibly more than tenfold below that of Buonanno et al.[75] This leads to emission rates of 0.01 quanta h$^{-1}$ for the case of a resting and only breathing occupant at their predicted 35th percentile[76]. On the other hand, Li et al.[77] estimated the median quanta emission rate in their study to be between 20 quanta h$^{-1}$ and 454 quanta h$^{-1}$.

Mikszewski et al.[78] provided values within the wide range from 0.0058 quanta h$^{-1}$ up to 4300 quanta h$^{-1}$ depending on expiratory activity, activity level and percentile. The calculated median emission rate for a SARS-CoV-2 standing and speaking emitter was 2.7 quanta h$^{-1}$, while the 95th percentile was at 250 quanta h$^{-1}$. For a resting and oral breathing emitter the respective values were 0.55 quanta h$^{-1}$ and 52 quanta h$^{-1}$. However, their listed and referenced SARS-CoV-2 outbreak events show an above-average emission rate between the 73rd and 98th percentile of the standing and speaking emitter distribution (15 quanta h$^{-1}$ to 970 quanta h$^{-1}$) with one exception that is even outside the predicted range[78].

To conclude, the distribution assumed in this paper, although based on formerly proposed values, might be considered a rather conservative approximation with respect to a median emission scenario. On the other hand, this last limitation does not restrict the findings of this study, which result from the comparison of different cases given the same assumed distribution. It rather illustrates the case dependency of super-spreading events.

## Conclusion

Overall, the analytical approach proved to be suitable for the risk assessment of venues with MV. However, the observed sensitivity to boundary conditions limited its use, even for investigating the effects of different parameters and mitigation strategies. Therefore, an individual infection risk assessment through experimental and numerical approaches is required to cover various ventilation concepts and to identify venue-specific high-risk sites and areas of poor air circulation. The results of the study highlighted the wide distribution of individual infection risks. Low-, medium- and high-risk sites varied according to the ventilation strategy, the emitter position and the emission mode. All three ventilation strategies studied showed high-risk positions in the near field of the emitter, but further distribution in space was different. At all venues, high-risk positions were also observed well beyond the physical distance guidelines. Venues with DV had the lowest overall risk of infection and number of secondary cases with an $R_{CFD}$ value well below one, even when the Omicron variant was considered. The observed directional aerosol distribution allowed the prediction of highly exposed positions and the expected number of secondary cases per event. However, in unventilated areas, aerosols can accumulate and locally increase the risk of infection. In venues with MV or NV, predicting highly exposed positions is particularly difficult due to the influence of boundary conditions and room parameters (air inlets and outlets, windows, room height, volume) on the room airflow. Face masks provide the best protection against aerosol transmission but should be combined with other mitigation measures in high-risk areas and situations. In terms of pandemic preparedness, the connection of the stage area to the ventilation system should be enforced, as well as raising the awareness of stage technicians and directors of the benefits of a well-adjusted ventilation system in reducing the transmission risk. However, the airflow rate should be balanced between the maximum acceptable individual risk of infection and economically acceptable operating costs.

## Methods
### Study design
The airborne transmission risk potential of venues with different room characteristics and ventilation concepts was examined using three approaches: experimental measurements using the Aerosol Transmission Measurement System (ATMoS), Computational Fluid Dynamics (CFD) analyses and the analytical Wells-Riley model (Fig. S1). In all three approaches, one infectious person (emitter) was placed in a fully occupied audience. As recent studies have confirmed the presence of people with high viral loads, so-called high emitters[61,64,79,80], two emission profiles were considered: (I) a sedentary, passive emitter with an average viral load and (II) a slightly active emitter with a high viral load at the 90th percentile. We chose the 90th percentile as a moderate percentile for a high emission case to avoid overestimating the potential quanta emission given the uncertainties associated with high emission rates and long tail probability distribution functions. The experimentally and numerically derived absorbed aerosol or quanta concentrations were used to calculate the individual ($P$) and global risk of infection ($R$), an analogue of the event reproduction number and an estimate of the effect of a single infectious occupant at an event on virus transmission[81,82]. $R$ also represents the number of secondary infections caused by an infectious individual at an event[75]. Analogous to the basic reproduction number $R0$, an estimate of the virus transmissibility, $R$ should be kept at <1 to control disease transmission and epidemic growth[83]. Additionally, special cases were considered by all three approaches: (I) a venue combining displacement ventilation and natural ventilation (HVV), (II) an infectious actor and (III) varying boundary conditions including reduced airflow rate and checkerboard pattern seating. Furthermore, the effects of mitigation measures, virus variants and varying boundary conditions such as the use of face coverings, residence time, airflow rate and occupancy on the risk of infection were investigated using CFD results. The study design is shown schematically in Fig. S1.

### Venues
To cover commonly installed ventilation systems, venues with displacement ventilation (DVV), mixing ventilation (MVV) and natural ventilation (NVV) were selected. All the venues studied are theatres with an auditorium layout with ascending rows of seats, ranging from 99 to 470. Information on room characteristics and positions of air inlets and outlets are shown in Table S1 and Fig. S2.

### Experimental measurements
The experimental measurements were carried out with ATMoS and were performed as previously described[39,40]. In brief, ATMoS consists of an emitter, the Atomizer Aerosol Generator ATM 230 (Topas GmbH, Dresden, Germany), that continuously releases a 10%-NaCl-water solution into the environment with a mass flow of 0.43 g/min and an aerosol mean diameter of 2.4 μm with a standard deviation of 1.1 μm. After evaporation, the virus-sized NaCl nuclei remain in the air and follow the room airflow, thus serving as an ideal virus surrogate. Seven absorbers were distributed in the room, which inhaled the released aerosols at a flow rate of 10 l/min, generated by a vacuum pump (DC 12 V 12 W V, VN-C3 Mini, Vikye, China) with inhalation tube. A fine filter (original coffee filter 1×6, Melitta,

Germany) is attached to the end of the inhalation tube, surrounded by deionized water and acts as an atomizer, breaking down the air stream into fine bubbles. The absorbed particles were dissolved in ultrapure water and were quantified by conductivity measurement over time. Therefore, the conductivity sensor HI98192 (Hanna Instruments, Germany) with a resolution of 0.01 μS/cm and a measurement accuracy of $+-1\%$ was used. The experimental setup was as follows: 10 min lead-in time to measure the background concentration at each location (no aerosol emission), 27-60 min aerosol emission (Table S1) and 10 min lead-out time (no aerosol emission). For the determination of aerosol emission during regular events, the measurement duration was adapted to event-specific processes, such as the timing of half-times, resulting in different measurement periods. To simulate the influence of body-generated buoyancy effects on aerosol distribution and airflow characteristics, up to 100 heat sources were distributed throughout the venue mimicking a human heat emission of 80 W. In venues with more than 100 seats, experimental measurements were carried out during regular events with spectators and heat sources.

**Calculation of the experimental risk of infection ($R_{ATMoS}$).** Using the inhaled mass of NaCl, the absorber-specific inhaled quanta $D_q$ dose was calculated[37]:

$$D_q = \int_0^t \dot{q}_{in}(t)\,dt = \frac{\dot{q}_{out}}{\dot{m}_{out\ sp}} \int_0^t \dot{m}_{in\ sp}(t)\,dt \qquad (1)$$

with $\dot{q}_{in}, \dot{q}_{out}, \dot{m}_{out\ sp}, \dot{m}_{in\ sp}$ and $t$ as quanta input and output rate, mass flow for NaCl output and input and time. According to the Wells-Riley approach, the experimental individual infection risk via aerosols $P_{ATMoS}$ was calculated for each absorber as[84]:

$$P_{ATMoS} = 1 - e^{-D_q} \qquad (2)$$

A quanta emission rate of 18.6 quanta h$^{-1}$ was assumed for a sedentary, passive emitter and 265 quanta h$^{-1}$ for a high emitter (high emitter scenario)[29,75]. To calculate the venue-specific global infection risk $R_{ATMoS}$, the mean of the seven absorber-specific $P_{ATMoS}$ was calculated and multiplied by the maximum number of occupants (N) per venue as:

$$R_{ATMoS} = P_{ATMoS}\, N \qquad (3)$$

**Computational fluid dynamics (CFD)**
The presented CFD study includes four venues of particular interest: one displacement ventilation case (DVV), one multi-purpose mixing ventilation venue (MVV) and one ascending stage case with nonspecific ventilation concept (natural ventilation, NVV). Furthermore, a special case of displacement ventilation with ventilated stalls and two not-mechanically ventilated balconies (hybrid, HVV) was investigated. CFD analyses were conducted for varying boundary conditions such as occupancy (full vs. checkerboard) for DVV and MVV, airflow rates (100% vs. 50%) for DVV and emitter positions for DVV, HVV, MVV and NVV. Using the software Simcenter™ STAR-CCM+, steady-state simulations on unstructured finite volume grids are conducted after simplified but detailed reconstruction of the geometric features and boundary conditions based on construction plans, interviews with the responsible technical staff of the venues and inspection of the venue-specific conditions on-site including measurements of the thermal conditions, e.g. temperatures of the environment, the supply air and the surroundings. Flow and energy transport are solved in a segregated manner using the SIMPLE algorithm and the segregated fluid temperature model. Turbulence is modelled by the Realizable k-ε-Model in a Two-Layer formulation (Wolfstein). Room air is assumed to be a single component ideal gas under the influence of gravity. Considering the substantial heat fluxes of lighting, electrical devices and occupants, grey thermal surface-to-surface radiation is applied under usage of view factors. Computer simulated persons (CSP) depict simplified, seated occupants, where the mouth area is specifically distinguished for the insertion of breath tracer gases. CSP are assumed to emit a heat flux of 80 W. Each pre-selected emitter releases a personalized passive scalar with a fictitious, momentum-free mass flux through the mouth area surface cells, which is thereafter transported by convection and diffusion. The passive scalar values throughout the venue's volume can be referred to their respective source flux and thus local, non-interacting concentrations are obtained within each cell for each emitter. For each CSP, a hemisphere of radius 0.23 m around the mouth normal vector is defined, which acts as a volume-averaged sampling zone assigned to the respective absorbing CSP. The averaged values approximately represent the locally experienced, relative amounts of aerosols shed by the different emitters. These values, along with additional information for further analysis and normalization, are exported and subsequently evaluated in tailored Python scripts. This approach also allows for an a posteriori assignation of the typically uncertain quanta emission rate.

Base sizes of the grid range from 0.1 m to 0.3 m, depending on the size of the venue. Typically, cell sizes are much smaller and rather on the order of a few centimetres in the proximity of CSP, furnishings or equipment. Local refinements, especially on heat or passive scalar emitting surfaces like the CSP, are on the order of millimetres and prism layer cells (4 to 6 layers regularly) support the near-wall solution. Overall mesh sizes range from 3.5 million to 34 million cells. Solution of the flow variables is performed first, while the passive scalar transport equations are solved on the frozen flow field afterwards. Convergence is assumed on the basis of a relative residual drop of at least three orders while simultaneously ensuring constant and physically reasonable monitor values for relevant integral values of the solver variables, e.g., temperature or passive scalar fluxes.

For validation, we used two sets of measurement data (B3, B4) provided by Li et al.[85,86]. They investigated the combined effects of convection, conduction and radiation in a full-scale room model for displacement ventilation. Since their boundary conditions, case of application and physical effects of interest are quite similar, this study appeared as a reasonable benchmark. We applied our models, e.g., turbulence and wall treatment models, radiation models and meshing approach to two of the given test cases with according boundary conditions. The thermocouple measurements and the according values extracted from our CFD prediction are in good agreement as shown in Fig. S9.

**Calculation of the numerical risk of infection ($R_{CFD}$).** To the present day, there is still uncertainty concerning the quanta emission rate of SARS-CoV-2 aerosols. Peng et al.[29] established an approximately log-normally distributed emission rate with a mean of 18.6 quanta h$^{-1}$ for a sedentary, passive emitter of the wild-type variant, where the 5th and 95th percentile are located at 8.4 and 48.1 quanta h$^{-1}$, respectively. Buonanno et al.[75] specify comparable log-normally distributed emission rates for different activity and vocalization levels. Between the two studies the deviations in reported mean values and standard deviations are partially balanced by enhancement factors to compensate for case differentiation. There is no log-normal distribution which fulfils all three requirements stated above for the mean and the two given percentiles. Moreover, non-passive behaviour, e.g. (quiet) speaking, is not incorporated in the distribution of Peng et al.[29]. Since the emission profile of occupants is subject to personal variations and behaviour, we assume a combined log-normal distribution with mean value of 18.6 quanta h$^{-1}$ and standard deviation of $\sigma = 0.720 * \ln(10) \approx 1.65786$, where the latter is as suggested by Buonanno et al.[75]. Thus, the distribution to fulfil these conditions is given by LN($\mu,\sigma^2$) where the desired normal mean is given by $\mu = 0.672683 * \ln(10) \approx 1.54891$.

500.000 pseudo-random number realizations of this distribution have been computed to account for the variability of the quanta emission rate while the mean value of 18.6 quanta h$^{-1}$ was verified. Subsequently, for all realizations and all venues the corresponding quanta doses were calculated based on the locally experienced volume-averaged values within the CSP hemispheres and the event duration according to Eq. (1) (i.e., steady-state absorption is assumed). By applying Eq. (2) the local risk $P_{CFD}$ with respect to a given emission rate is evaluated. In a last step, the average of all realizations within a particular venue is calculated as $P_{CFD}$, creating a mapping of

the mean, local infection risks with regard to the given emission rate distribution. For further analysis, high emission cases without variation (265 quanta h$^{-1}$) are covered. Furthermore, an individual acceptable risk of infection $R_{acc}$ was determined for numerically derived infection risks and set at $10^{-2}$ in accordance with recent studies[75,87,88] as the acceptable level of the COVID-19 risk of infection is still unknown. When planning future events, the individual infection risk must be less than $R_{acc}$ to keep the risk of infection to spectators manageable. This enabled the identification of high-risk areas and risk management at the venues studied.

Furthermore, a risk analysis was conducted, considering the effect of mitigation measures, virus variants and varying boundary conditions. In detail, cases with different mask efficiencies (surgical mask: 65%, FFP2/N95 mask: 80%), SARS-CoV-2 virus variants (Alpha, Delta and Omicron with enhancement factors as in Table S2) as well as variations of duration (1, 2 and 3 h) and vocalization were compared by taking into account the multipliers of the quanta absorption (see Table S2).

### Modified Wells–Riley approach

The analytical Wells–Riley approach[84,89] was applied to each venue with modifications[29] to prove its applicability for a venue-specific infection risk assessment.

**Calculation of the analytical risk of infection ($R_{analyt}$).** For classification of venues in terms of their potential airborne infection risk, the risk parameter $H$ was introduced by Peng et al.[29] To account for the effectiveness of air distribution of the different ventilation systems and to improve the imperfect well-mixed assumption of the analytical model, the parameter ventilation effectiveness ($E_z$) was introduced into the equation of Peng et al.[29], similar to an approach of Sun & Zhai[69], by multiplying the air exchange rate (AER) $\lambda$ by $E_z$:

$$H = \frac{r_{ss} r_E r_B f_e f_i D N_{sus}}{V(\lambda E_z)} \tag{4}$$

where $r_{ss}$ is the correction factor for the deviation of average quanta concentration from that of steady state, e.g., for events too short to approximately reach steady state, $r_E$ is the activity-related shedding rate enhancement factor, $r_B$ is the activity-related breathing rate enhancement factor, $f_e$ and $f_i$ are the exhalation and inhalation penetration efficiency for face covering, $D$ is the duration of the event, $N_{sus}$ is the number of susceptible persons and $V$ is the indoor environment volume. Ventilation-specific values for $E_z$ can be found in the ASHRAE Standard 62.1[14] and are listed in Table S1 for the venues studied. In considering the worst-case scenario, virus decay and the deposition rate of virus-containing particles in the air were assumed to be low and therefore neglected. The analytically derived global risk of infection ($R_{analyt}$) was calculated with parameter-specific values analogous to Peng et al.[29] shown in Table S2:

$$R_{analyt} = E_{P0} B_0 I H \tag{5}$$

with $E_{P0}$ the SARS-CoV-2 exhalation rate of a resting and only breathing infector, $B_0$ the breathing rate of a resting susceptible person and $I$ the number of infectors present. The breathing rate was set to 0.49 m³/h. The basic configuration represents a typical pre-pandemic event in different venues with the presence of one infectious person and was defined as follows:

- duration of the event: 2 h
- occupancy: 100%
- activity level: sedentary, passive
- SARS-CoV-2 variant: wild-type
- no face coverings

For surgical masks, a penetration efficiency of 0.35 (65%) was assumed, i.e. 35% of exhaled particles still pass through the mask when both infectious (mask exhalation efficiency 50% (0.5)) and susceptible persons (mask inhalation efficiency 30% (0.7)) wear a mask. A combined filtration efficiency of 96% (0.04) was assumed for well-fitting FFP2/N95 masks worn by emitting and susceptible persons.

## Data availability

The data that support the findings of this study are available from the corresponding author upon reasonable request.

## Code availability

The codes that were used for this study are available from the corresponding author upon reasonable request.

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

## Acknowledgements

This work was supported by the Ministry of Science, Energy, Climate Protection and Environment of the Federal State of Saxony-Anhalt (grant number I 140), the Federal Government Commissioner for Culture and the Media (grant number 2521NSK115) and Berlin University Alliance (grant number SARS-CoV-2 Ausbreitung, Restart 2.0) as part of the RESTART 2.0 project. We especially thank the venue managers, the stage technicians and the staff of the Puppet theatre Halle, the New Theatre Halle, Opera Halle, the Maxim-Gorki Theatre Berlin and Volksbühne Berlin for the opportunity to carry out the experimental measurements with ATMoS, for their support and for providing floor plans and technical drawings for CFD analyses. We would like to thank Ken J. Lindenberg and Anastasia Strigaleva for their assistance with the CFD analysis and Sophia Wolff and Katharina Schmidt for their assistance with the experimental measurements.

## Author contributions

S.M.G.: conceptualization, methodology, formal analysis, investigation, resources, data curation, writing—original draft, writing—review and editing, supervision, visualization, funding acquisition. K.H.L.: conceptualization, methodology, software, validation, formal analysis, investigation, resources, data curation, writing—original draft, writing—review and editing, visualization. F.H.: conceptualization, methodology, validation, formal analysis, investigation, resources, data curation, writing—review and editing, supervision. I.S.: conceptualization, methodology, validation, formal analysis, investigation, resources, data curation, writing—review and editing. U.K.: conceptualization, methodology, resources, writing—review and editing, supervision. M.K.: conceptualization, methodology, res ources, writing—review and editing, supervision. O.P.: conceptualization, methodology, resources, supervision, funding acquisition. S.S.: conceptualization, methodology, investigation, resources, writing—review and editing. Ü.H.: conceptualization, methodology, formal analysis, investigation, resources. G.B.: methodology, software, formal analysis, investigation, resources, data curation. A.M.: conceptualization, supervision. K.S.: formal analysis, investigation. S.M.: conceptualization, methodology, investigation, resources, writing—review and editing, supervision, project administration, funding acquisition.

## Funding

## Competing interests

The authors declare no competing interests.

## Ethics

The Ethics Committee of the Martin Luther University (Halle, Germany) approved the aerosol measurement experiments with the ATMoS system during regular events with spectators.
