## [Peer Review File · Communications Engineering]

This manuscript has been previously reviewed at another journal. This document only contains reviewer comments, rebuttal and decision letters for versions considered at Communications Engineering.

SARS-CoV-2 airborne infection risk in tiered auditorium venues: comparison of mitigation strategies using experimental, numerical and analytical approaches

Corresponding Author: Dr Sophia Mareike Geisler

Version 0:

Reviewer comments:

Reviewer #1

(Remarks to the Author)

The article claims to assess the airborne transmission risk of SARS-CoV-2 in venues with different ventilation strategies using experimental, numerical and analytical approaches, which it ultimately does achieve. The questions arise on the verification of the methodology of the obtained results.

The approach is novel in scale and application, as well as in combination of 3 methodologies.

The paper will be of interest to others in the field, and it could influence the thinking as it considers different approaches to infection risk estimations.

The manuscript is clearly written and reads well. The manuscript is of good length and it communicates important findings clearly.

Recommendations and questions:

- It would be good to further discuss the quanta emission rate effect on the results of the study. What are the uncertainties with selecting this quanta emission rate?
- Does CFD take into account the evaporation of aerosols for example and how would this affect the results?
- Previous literature has been sufficiently reviewed to identify assumptions and uncertainties in previous studies.
- The CFD risk calculated claims to be in good agreement with experimental and analytical values- this is not supported by the results.

Resubmission is highly encouraged after proof of CFD approach verification is included. The current version of the manuscript "validates" CFD calculated risk with analytical and experimental validated risk. Turbulence might play a factor so it's relevant to validate the CFD with airflow/temperature measurements for different ventilation strategies.

Reviewer #2

(Remarks to the Author)

The paper under review is a useful and novel work that compares the risk of infection between different modes of ventilation. It further compares the measurements with CFD and analytical methods. There are a few concerns which the authors need to address before I can recommend acceptance for publication.

In my understanding if the measured data from Atmos are considered ground truth, the average errors produced by CFD and Analyt are not much different – this is not surprising given the modeling assumptions introduced. A systematic error analysis should be performed. In my opinion the relevant Error could be defined as the absolute difference between R_{ATMOS} and R_{CFD} or the absolute difference between R_{ATMOS} and R_{ANALYT} (see attached table), because this is what matters; a percentage error seems to bias the error towards unimportant small R_{Atmos} values which being less than 1 is not of interest anyways.

Line 149: Why choose the 90th percentile? A recent modeling study suggests that 80% of infections are caused by the top 4% percentile high emitters. <https://pubs.aip.org/aip/pof/article/34/5/051914/2846582> In the context of Fig. 4 it would be interesting to see how much reduction in R is caused by a 50% reduction in airflow rate high emitters.

Reviewer #3

(Remarks to the Author)
Paper review

This paper combines measurements and modeling to better understand what variables drive the risk of COVID-19 infection in a theater. I appreciate the comprehensive use of measurements, CFD, and numerical modeling. The paper is a nice addition to the literature on airborne infectious disease mitigation strategies and should be published. I recommend some edits to help make the paper more readable. As it is, it is often difficult to follow, with results as lists of risk percentages as a function of scenario.

I recommend that the title be updated so that it is clear that this is a study of a theater. There is no discussion in this paper of how applicable the results would be to other sorts of public buildings. So a suggested new title is "SARS-CoV-2 airborne infection risk in amphitheaters with different mitigation strategies – a comparison between experimental, numerical and analytical approaches." I also think you can broaden it to include mitigation strategies generally since the study investigates all the available tools, including masks, distancing, reducing occupancy, ventilation...

I recommend that you discuss briefly with citations why DV has not been widely implemented in buildings. From the results of your study, it would seem that DV should be reconsidered as a ventilation strategy to better reduce infectious disease transmission. However, in the assessment of DV in the previous decades of ventilation design, it was found to cause some problems with indoor air quality and moisture, etc. so that it was in the end not used, and mixing ventilation was preferred.

Introduction

Line 47. A statement that COVID-19 is transmitted less frequently through direct contact or fomites has a citation of WHO and CDC documents. I do not think that these documents provide evidence that transmission happens through direct contact or fomites, so please update your citations to provide references that show this transmission route has been documented or delete this statement.

Line 51: airborne aerosols should either be aerosols or airborne particles.

Line 52: what hygiene concepts are you referring to? Be more specific because if you are referring to disinfecting surfaces, then there is no evidence that this keeps the risk of COVID-19 infection low. If you are speaking more generally of strategies to reduce aerosol concentrations, then use a better phrase like "appropriate mitigation strategies that reduce exposure to infectious aerosol"

Line 97 and line 99: here you use the phrase "assess the airborne transmission risk two times. Please rewrite so that you only use this phrase one time.

Line 100: change the abbreviations of your scenarios to DV, MV, and NV. There is no reason to have 2 "V" in the abbreviation.

Results

Line 114: what does the "-1" and "-2" mean here? Since the paper is written so that the results are first before the methods, you must explain briefly. I recommend stating that you are analyzing 4 different ventilation scenarios. One each DV, MV, and NV and a hybrid of DV and NV. You could use the abbreviation for this special case as DV+NV.

In the first paragraph of your results please describe how generalizable these results are. I do not think there are so clear here that you are only assessing what type of ventilation would be better for lowering infectious disease risk in an amphitheater-type building.

Line 180-181: the sentence "However, all derived values were lower than R_{analyt} " is confusing. Do you mean that the risk of infection predicted by CFD and ATMoS was lower than the risk from the analytical model? Please state this more clearly.

Line 205 - what does the abbreviation FFP2 mean? Spell out and why do you need a number "2" here?

Line 215: why did you only assess the impact of reducing the airflow rate for DV with CFD? The beauty of a CFD model is that it is much easier to assess all sorts of different airflow rates, so you also need to assess the impact of reducing the airflow rate for MV too or explain why you didn't.

Line 234: it is unclear whether the sentence starting with "Numerical..." refers to the CFD modeling. Please be consistent with your terminology and if you are going to use CFD versus analytical in the paper, keep to this terminology to avoid confusion.

Line 240: when you state that there was a "2.7 times higher risk of infection compared to the stalls..." are you referring to the balcony? Please be clear and state this.

Line 256: try to reduce the complexity of your sentences so for example in this line, it is fine to delete the phrase “which is 2.7 times lower compared to emitter position B9.” Since the numbers are provided it is clear it is much lower.

Line 381: change Most to Many

Lin 405: your crowding index did not result in much risk reduction. I recommend finding out using your modeling what crowding index would make an impact. That is a much more useful result, since reducing occupancy was used a lot to reduce transmission and many studies suggested that this was an effective strategy.

Line 420, be clear that the reason that the modified analytical Wells-Riley approach did not represent a risk as well is because of the directional flow patterns resulting from the ventilation. This in turn caused spatio-temporal differences in risk.

Methods

one issue with the methods is that there is no assessment of uncertainty in the measurements or sensitivity to the parameters used in the models. I recommend that the authors provide an assessment of uncertainty in their predictions of risk.

Line 513: the term “absorbers” is not very scientific and you should use standard terminology for aerosol measurement methods. Are you using impingers to collect the aerosol? open face filters? Provide the make and manufacturer here.

Line 515: similar comment on the conductivity measurement - what instrument is this?

Line 529. Why do you have 5 significant figures for your quanta emission rate? Better to use 264 q/h

Line 625: "an" should be "a"

Signed by Professor Shelly L. Miller, University of Colorado Boulder, department of Mechanical Engineering, USA

Version 1:

Reviewer comments:

Reviewer #1

(Remarks to the Author)

Thank you for addressing my comments, I am content that concerns I had with the original version of the manuscript have now been addressed. In my opinion, the manuscript should be accepted for publication.

Reviewer #3

(Remarks to the Author)

The authors did an excellent job responding to reviewer comments. I am very impressed and appreciated their approach and their hard work to improve the paper. I do not have additional comments and I recommend publication.

Response to referees

Reviewer #1

1. It would be good to further discuss the quanta emission rate effect on the results of the study. What are the uncertainties with selecting this quanta emission rate?

We appreciate your comment and agree with your request to include a critical discussion of the uncertainties of selected quanta emission rates. We have included an extended discussion of this issue in the Discussion section. Please, see line 430-454, shortly before the Conclusion section, where we try to outline the range of possible quanta emission rates and their effect on our results.

2. Does CFD take into account the evaporation of aerosols for example and how would this affect the results?

Unfortunately, CFD results do not take into account the evaporation effects. Also, the deposition of the emitted aerosol is neglected. These are indeed limitations of our study and we will point out these more clearly in our discussion part (line 425-429):

“From a modelling perspective, the lack of evaporation and deposition effects in the CFD model is a limitation regarding general aerosol droplets. As a simplification, we imply for our aerosol to be perfectly airborne (in experiments and simulations), which may be partly justified by the aerosol size distribution measured in Schulz et al.⁴² Arguably, the consideration of aerosol deposition could perhaps close the gap between the experimentally measured and simulated concentrations to some extent.”

We assume that the inclusion of aerosol evaporation would, in general, lead to a more complex, size-dependent outcome for aerosol displacement as described for example in Yan et al. . As a simplification, we imply for our aerosol to be perfectly airborne (in experiments and simulations), which may be partly justified by the aerosol size distribution measured in Schulz et al.⁴²: (“mean diameter of 2.4 μm with a standard deviation of 1.1 μm ”). Essentially, the effect of evaporation would take place within the first few seconds after exhalation (for smaller droplets) so that a mostly passive transport of the droplet nuclei in the size range of interest appears as a reasonable approximation. Arguably, the consideration of aerosol deposition could perhaps close the gap between the experimentally measured and simulated concentrations to some extent.

References

Schulz et al. (2023) Experimental device to evaluate aerosol dispersion in venues. Preprint at medRxiv. Doi: 10.1101/2023.06.13.23291335

Yan et al. (2019) Thermal effect of human body on cough droplets evaporation and dispersion in an enclosed space. *Build Environ.* 148: 96-106

3. The CFD risk calculated claims to be in good agreement with experimental and analytical values- this is not supported by the results.

We assume that you refer to the results shown in Table 1 and described in section “*Comparison of the experimental (R_{ATMoS}), numerical (R_{CFD}) and analytical (R_{analyt}) derived risk of infection for different ventilated venues*”. Perhaps, we should be more precise in our wording. We try to express that the agreement for DVV between R_{ATMoS} and R_{CFD} is good for the emitter positions E2, E6 and H5. There is a clear deviation to R_{analyt} for all 4 positions. Furthermore, would you agree, if we describe the assessment for the MVV results as to be in reasonable agreement between R_{ATMoS} , R_{CFD} and R_{analyt} ? For case NVV, we may change to the following (line 153-159):

“For the NVV position A14, the values for R_{ATMoS} , R_{CFD} and R_{analyt} were all markedly higher than for the other types of ventilation. However, while the risk assessment based on analytical values

revealed (more than) a doubling of the values compared to MVV, the experimental and CFD values showed a (more than) 5-fold or (close to) 7-fold increase, respectively. Therefore, the agreement for NVV between the three approaches is worse than for DVV and MVV. A reason for this observation might be given by the steady state correction factor r_{ss} given in Peng et al.³⁰, which is used for R_{analyt} but not for R_{CFD} .”

4. Resubmission is highly encouraged after proof of CFD approach verification is included. The current version of the manuscript “validates” CFD calculated risk with analytical and experimental validated risk. Turbulence might play a factor so it’s relevant to validate the CFD with airflow/temperature measurements for different ventilation strategies.

Thank you for your encouragement which we highly appreciate. Yes, you are right. During most of the events at the venues, surface and room temperatures were measured. These were mostly in good agreement with the results of the subsequent CFD simulations. Flow field validation was a difficult task due to the highly complex setups and the participation of many attendees in some cases. A qualitative validation of the modelled flow field by means of smoke visualization of the thermal plumes was done at the beginning of the project. Unfortunately, quantitative validation data for the real venue flow field apart from the mentioned is not at hand. Therefore, we decided to include a validation study based on the measurement data of Li, Sandberg and Fuchs (1992, 1993a). This case was also studied by means of numerical simulations before (Li et al. 1993b, Gilani et al. 2016). We applied our models, e.g., turbulence and wall treatment models, radiation models and meshing approach to two of the given test cases with according boundary conditions. The studied case used a similar problem as given for our DVV cases (displacement ventilation, buoyancy, consideration of radiation and convection as far as possible). As shown in the figures below, the results for the temperature values along the measurement pole are in good agreement (for case B3). The agreement for the velocity values is mostly acceptable or good at some locations. However, the decay of momentum in the upper half is not depicted as desired. However, former simulation studies struggled even more so in capturing these effects.

In the text we have added the following paragraph (line 577-583):

“For validation, we used two sets of measurement data (B3, B4) provided by Li et al. (1992, 1993a) They investigated the combined effects of convection, conduction and radiation in a full-scale room model for displacement ventilation. Since their boundary conditions, case of application and physical effects of interest are quite similar, this study appeared as a reasonable benchmark. We applied our models, e.g., turbulence and wall treatment models, radiation models and meshing approach to two of the given test cases with according boundary conditions. The thermocouple measurements and the according values extracted from our CFD prediction are in good agreement as shown in Figure S9.”

References

- Gilani et al. 2016. CFD simulation of stratified indoor environment in displacement ventilation: Validation and sensitivity analysis. *Build Environ.* 95: 299-313
- Li et al. (1992). Vertical Temperature Profiles in Rooms Ventilated by Displacement: Full-Scale Measurement and Nodal Modelling. *Indoor Air.* 2(4): 225-243
- Li et al. (1993a) Effects of thermal radiation on airflow with displacement ventilation: an experimental investigation. *Energy Build.* 19(4): 263-274
- Li et al. (1993b) Numerical prediction of airflow and heat-radiation interaction in a room with displacement ventilation. *Energy Build.* 20(1): 27-43

Reviewer #2

1. In my understanding if the measured data from Atmos are considered ground truth, the average errors produced by CFD and Analyt are not much different – this is not surprising given the modeling assumptions introduced. A systematic error analysis should be performed. In my opinion the relevant Error could be defined as the absolute difference between R_{ATMOS} and R_{CFD} or the absolute difference between R_{ATMOS} and R_{ANALYT} (see attached table), because this is what matters; a percentage error seems to bias the error towards unimportant small R_{Atmos} values which being less than 1 is not of interest anyways.

We appreciate your very helpful remark that hits an important point! The experimental and numerical group discussed the same topic at multiple instances. Unfortunately, we arrived at a different opinion in the end. One problem is that the evaluation based on absolute values would bias the results towards the major influence of NVV (which also has the least amount of measurement data). The differences between R_{CFD} and R_{Analyt} observed for NVV are most likely particularly high because of a correction factor r_{ss} as proposed by Peng et al. for cases, where steady state concentration is not reached. This is clearly a drawback of the simulation approach. But the experimental group also hesitates to state their measurements as ground truth, not only because of the limited number of measurement positions but also because of the intermittent conditions experienced during the measurements/events which were not at all perfect laboratory conditions. This was also highlighted in the discussion part (see lines 395-404). In consequence, to our understanding it is rather the joint consideration of all three approaches

that gives a more complete picture. The results (especially of MVV and DVV) thereby depict in some way the range of values that is to be expected given that unpredictable influences might disturb the carefully selected boundary conditions in a real-life scenario. For a systematic error analysis, we hope to have a much richer set of measurement values in future studies, in order to create more meaningful assessments.

References

Peng et al. (2022) Practical indicators for risk of airborne transmission in shared indoor environments and their application to covid-19 outbreaks. *Environ. Sci. Technol.* 56: 1125–1137.

2. Line 149: Why choose the 90th percentile? A recent modeling study suggests that 80% of infections are caused by the top 4% percentile high emitters. <https://pubs.aip.org/aip/pof/article/34/5/051914/2846582> In the context of Fig. 4 it would be interesting to see how much reduction in R is caused by a 50% reduction in airflow rate high emitters.

Thank you for this helpful reference and your additional information. Certainly, the choice of the 90th percentile is arbitrary to some degree. The basis for this choice is within the study of Buonanno et al.⁷⁶ Firstly, the authors state that:

“The individual infection risk ($R(ER_q)$) presents a maximum value at the 92nd percentile of ER_q of a singer/speaker (i.e. about 330 quanta h^{-1}).”

They also point out that:

“Due to the similarity of the probability density functions of the four expiration activities resulting from the calculation of the quanta emission rates ($\log_{10}(ER_q)$ reported in Table 2), the pdf_R for all the exposure scenarios tested here were similar to that of the exposure scenario shown in Fig. 2 (i.e. the maximum $R(ER_q)$ values occur in the narrow range of 90th-95th percentile).”

Secondly, they give the quanta emission rates for the 90th percentile for reference (but also for the 95th). In order not to overestimate the potential quanta emission and select a reasonable value, given the uncertainty that still remained, we opted for the lower percentile. In the introduction of the reference you shared, the authors refer to another study (Chen et al. (2021), which links to three more studies) where the authors state that:

“such superspreading events are characterized by overdispersion in SARS-CoV-2 transmissions with 10%–20% of index cases responsible for 80% of secondary cases.”

Given these data points, the 90th percentile approximation for a high emitting case might be justifiable, although it appears that consensus has not been reached. The problem may be of a more general nature considering the long tail probability distribution functions.

Concerning your second question, we are not quite sure, but we assume that you refer to the 50% reduction of exhalation air flow while keeping the air change rate constant. Since respiratory rate was set fixed for our study, we did not include this in the article. However, the effect would result in a halving of the emitted quanta which is comparable to a halving of the inhaled quanta dose of the occupants. This effect is indirectly and partly covered in Table 2 of the manuscript where the reduction of the time to 1h is listed.

References

Buonanno et al. (2020) Quantitative assessment of the risk of airborne transmission of SARS-CoV-2 infection: prospective and retrospective applications. *Environ. Int.* 145: 106112.

Chen et al. (2021). Heterogeneity in transmissibility and shedding SARS-CoV-2 via droplets and aerosols. *Elife.* 10:e65774.

Reviewer #3

1. I recommend some edits to help make the paper more readable. As it is, it is often difficult to follow, with results as lists of risk percentages as a function of scenario.

We thank you for your comment and apologise for the inconvenience. We have edited the manuscript to make it more readable. We have tried to shorten sentences and be more consistent with our wording.

2. I recommend that the title be updated so that it is clear that this is a study of a theater. There is no discussion in this paper of how applicable the results would be to other sorts of public buildings. So, a suggested new title is “SARS-CoV-2 airborne infection risk in amphitheatres with different mitigation strategies – a comparison between experimental, numerical and analytical approaches.” I also think you can broaden it to include mitigation strategies generally since the study investigates all the available tools, including masks, distancing, reducing occupancy, ventilation...

We appreciate your comment and agree that the manuscript lacks a discussion of applicability to other public buildings. Typically, amphitheatres are open-air circular or elliptical venues, often also used for sports or fighting events. The first component of the Greek word explicitly refers to “both sided” or “all around” (in contrast to a standard theatre). Unfortunately, this does not describe our typical arrangement. Since we were also looking for a short and precise formulation and already discussed a bit, we suggest changing “venue” to “tiered auditorium venue”. We also broaden the title to include mitigating strategies in general (changes in red):

“SARS-CoV-2 airborne infection risk in tiered auditorium venues: comparison of mitigation strategies using experimental, numerical and analytical approaches”

3. I recommend that you discuss briefly with citations why DV has not been widely implemented in buildings. From the results of your study, it would seem that DV should be reconsidered as a ventilation strategy to better reduce infectious disease transmission. However, in the assessment of DV in the previous decades of ventilation design, it was found to cause some problems with indoor air quality and moisture, etc. so that it was in the end not used, and mixing ventilation was preferred.

Thank you for this observation. Possibly, we phrase the results as too positive for displacement ventilation settings and we will consider a more differentiated phrasing. Indeed, for venues comparable to theatres or auditoria displacement ventilation might be preferable which explains the common application for these venues (see e.g., Siebler et al.(2023)). To the authors' knowledge, the main reasons for not considering displacement ventilation are based on practical problems during the planning and construction process rather than on any air quality, health or comfort considerations, which are (or were) unfortunately dismissed at the early stages of realization. However, there are still problems concerning indoor air quality discussed in the literature. We will refer to a selection of these for the sake of brevity with the following remark in the Discussion section (line 300-307):

“Although the results of this study appear favourable towards DV systems, and despite being often applied in practice⁵² (Siebler et al. 2023), a general recommendation of DV is not expedient. Bjørn and Nielsen⁵³ (2002) discuss the influential factors on contaminant distribution as e.g., motion, temperature gradient and mouth or nose exhalation. Similarly, Yuan et al.⁵⁴ point out, among other factors, limitations of DV with regards to the cooling load, space height, the wall (temperature) characteristic and the thermal comfort due to the vertical air temperature gradient. Additionally, Riffat et al.⁵⁵ (2004) emphasize the importance of an appropriate system design to avoid mixing polluted plumes and air in the occupied zone.”

References

Bjørn and Nielsen. (2002). Dispersal of exhaled air and personal exposure in displacement ventilated rooms. *Indoor Air*. 12(3): 147-164

Riffat et al. (2004). Review of research into and application of chilled ceilings and displacement ventilation systems in Europe. 28(3): 257-286

Siebler et al. (2023) A coupled experimental and statistical approach for an assessment of SARS-CoV-2 infection risk at indoor event locations. *BMC Public Health*. 23: 1394.

Yuan et al. (1998) A critical review of displacement ventilation. *ASHRAE Transactions*. 4101: 78-90

4. Line 47. A statement that COVID-19 is transmitted less frequently through direct contact or fomites has a citation of WHO and CDC documents. I do not think that these documents provide evidence that transmission happens through direct contact or fomites, so please update your citations to provide references that show this transmission route has been documented or delete this statement.

We appreciate your comment and have updated references 1 and 2. We have referenced two reviews that highlight the primary route of transmission via aerosols and point out the low transmission risk and epidemiological relevance of COVID-19 transmission via direct contact or fomites. We rephrased the sentence to make clear the low transmission risk of direct contact or fomites as follows (changes in red; line 17-19):

“The severe acute respiratory syndrome coronavirus type 2 (SARS-CoV-2) is the causative agent of the coronavirus disease 2019 (COVID-19) and is transmitted primarily by infectious respiratory droplets and aerosols; **alternative transmission pathways** through direct contact or fomites **can be considered of low epidemiological relevance**¹⁻²”

5. Line 51: airborne aerosols should either be aerosols or airborne particles.

Thank you, for your comment. We changed the word “airborne” to “infectious”. The sentence reads as follows (line 22-23):

“In fact, there are many reports of transmission events in confined and poorly ventilated indoor spaces, partly due to infectious aerosols.”

6. Line 52: what hygiene concepts are you referring to? Be more specific because if you are referring to disinfecting surfaces, then there is no evidence that this keeps the risk of COVID-19 infection low. If you are speaking more generally of strategies to reduce aerosol concentrations, then use a better phrase like “appropriate mitigation strategies that reduce exposure to infectious aerosol”

We are grateful for the comment, as the sentence is not appropriate as you have commented. We are speaking more generally of mitigation strategies to reduce aerosol concentrations. We have therefore rephrased the sentence as recommended (line 23-25):

“However, recent studies have shown that the event-related risk of contracting SARS-CoV-2 can kept very low with well-functioning ventilation systems and appropriate mitigation strategies to reduce exposure to infectious aerosols.”

7. Line 97 and line 99: here you use the phrase “assess the airborne transmission risk two times. Please rewrite so that you only use this phrase one time.

Thank you for your comment. We delete the second phrase (line 70-73):

“Therefore, ATMoS and CFD analyses were used in four different venues, one with displacement ventilation (DVV), one with mixing ventilation (MVV), one with natural ventilation (NVV) and one with a hybrid ventilation strategy combining displacement and natural ventilation (HVV).”

8. Line 100: change the abbreviations of your scenarios to DV, MV, and NV. There is no reason to have 2 “V” in the abbreviation.

Thank you for your comment. In lines 26 and 27 we introduced the general abbreviations for displacement ventilation (DV), mixed ventilation (MV) and natural ventilation (NV). To differentiate from these general concepts, we introduced additional abbreviations for the venues studied (the second 'V' stands for venue: DVV = displacement ventilation venue). For clarity and ease of understanding, we have decided to keep the abbreviations as DVV, NVV and MVV. We have followed your recommendation to abbreviate the hybrid ventilation differently, but we have decided to use HVV (Hybrid Ventilation Venue) for clarity instead of DVV+NVV, which could be misleading.

9. Line 114: what does the “-1” and “-2” mean here? Since the paper is written so that the results are first before the methods, you must explain briefly. I recommend stating that you are analyzing 4 different ventilation scenarios. One each DV, MV, and NV and a hybrid of DV and NV. You could use the abbreviation for this special case as DV+NV.

We appreciate and agree with your comment. We have removed the numbers in the abbreviations throughout the manuscript, except for MVV2 in the results (Checkerboard pattern seating) and supplements to distinguish between these two MV venues. See also comment on point 8.

10. In the first paragraph of your results please describe how generalizable these results are. I do not think there are so clear here that you are only assessing what type of ventilation would be better for lowering infectious disease risk in an amphitheater-type building.

Thank you for your suggestion! For improved clarity and to avoid overgeneralization, we will emphasize that we are referring to a specific type of venue by adding the following (changes in red; line 85-87):

"To obtain aerosol distribution data for the entire location and for every position in the audience, CFD analyses were performed for four different ventilation scenarios in multi-tiered, seated indoor event locations (DVV, MVV, NVV and HVV)."

11. Line 180-181: the sentence “However, all derived values were lower than R_{analyt} ” is confusing. Do you mean that the risk of infection predicted by CFD and ATMoS was lower than the risk from the analytical model? Please state this more clearly.

Thank you for your comment, and we agree that the sentence is unclear. We rephrase the sentence as follows (line 151-152):

“ R_{analyt} yielded the highest risk of infection compared to R_{ATMoS} and R_{CFD} .”

12. Line 205 - what does the abbreviation FFP2 mean? Spell out and why do you need a number “2” here?

Thank you for your comment. FFP stands for filtering facepiece and covers the nose, mouth and chin available with or without inhalation and/or exhalation valves. FFP masks must comply with the European standard EN 149, which defines three classes of such particle respirators, called FFP1, FFP2 and FFP3, according to their filtering efficiency. FFP2 masks filter at least 94% of airborne particles and have performance requirements, similar to N95 (United States) and KN95 (China).

To make the manuscript easier to understand for readers around the world, we used the term FFP2/N95 throughout the manuscript, following several other publications.

13. Line 215: why did you only assess the impact of reducing the airflow rate for DV with CFD? The beauty of a CFD model is that it is much easier to assess all sorts of different airflow rates, so you also need to assess the impact of reducing the airflow rate for MV too or explain why you didn't.

Thank you for your remark. First of all, we would like to mention the objectives of our study: We performed experimental and numerical measurements for the same venues in order to compare the results. Therefore, the numerical results were also dependent on the obstacles that appeared during the experimental measurements (see point b)) and not every measurement that can be realized by CFD can also be realized with experimental measurements in real-life scenarios. However, we agree that the beauty of CFD is that it allows for more and quicker parameter variation. To answer your question, we want to give 4 partly separated reasons.

- a) Indeed, we executed a MVV simulation with reduced airflow (10%, unfortunately, not the desired 50%). One reason was given by the observed high temperatures during the experimental measurements with ATMoS, so that we started to question the volume flow rate stated by the stage manager. However, the exploratory simulations showed good agreement for the observed temperatures with respect to the simulations at 100% volume flow rate. Temperatures received by CFD simulations were usually above 50 °C for reduced air flow rate (10%), as also was expected. Even with a volume flow rate of 50%, the anticipated local temperatures would be far beyond comfort level in this specific venue (more than 18 K temperature difference with respect to inlet temperature). Therefore, the results would not be realistically applicable and cannot be matched with experimental measurements.
- b) An objection to a) might be that a doubled airflow would have been reasonable in that case. However, here the problem was, as far as I can remember, that the venue was not able of further adjusting the volume flow rate, so that a comparative measurement would not have been possible in any case. Therefore, further numerical experiments with this parameter were stopped and disregarded for MVV.
- c) Due to time and financial constraints, a CFD simulation with 50% reduced airflow rates for MVV was not performed. This was also due to the fact that, as mentioned in b), the stage managers were unable to reduce the airflow rate during the experimental measurements. Furthermore, over the course of the whole project (2 years), we conducted experiments in 10 (or more) rooms and run simulations for 5 (or more, depending on the inclusion of rooms in the project). A total of approximately 1.5 TB of simulation data and more than 50 simulations accumulated. Several exploratory simulations for sensitivity tests, tests for boundary conditions, turbulence models and other modelling approaches remain partly uncounted. After 2 years, the resources for computing time and working time were consumed, much to our regret.
- d) As a reply to c), one could argue that the further results should also be published. However, we had to decide for the most essential findings and already exhausted word count and number of figures. Therefore, some results were disregarded completely from the manuscript.

14. Line 234: it is unclear whether the sentence starting with “Numerical...” refers to the CFD modeling. Please be consistent with your terminology and if you are going to use CFD versus analytical in the paper, keep to this terminology to avoid confusion.

Thank you for your comment, and we agree with the inconsistency you point out. In the introductory part, line 73, we introduced the term numerical and linked it to CFD. However, we have rephrased the sentence to which you refer (changes in red; line 209-211):

“CFD analyses (Figure 3, Fig. S3C) and experimental (Fig. S4) measurements observed a small aerosol plume with increased exposure behind the emitter for the position R8S21.”

15. Line 240: when you state that there was a “2.7 times higher risk of infection compared to the stalls...” are you referring to the balcony? Please be clear and state this.

We appreciate your comment, and we agree that the sentence is unclear. The derived numerical risk of infection indicates an overall risk of infection for the entire venue. The phrase “compared to the stalls” was used for the scenario where the emitter is placed in the stalls. To be clear, we rephrased the sentence for better comprehension (changes in red; line 213-216):

“On the contrary, aerosol emissions emanating from the balconies vanished slowly without exposing the stalls, but showed a 2.1 to 2.7 times higher risk of infection compared to the scenario where the emitter was placed in the stalls.”

16. Line 256: try to reduce the complexity of your sentences so for example in this line, it is fine to delete the phrase “which is 2.7 times lower compared to emitter position B9.” Since the numbers are provided it is clear it is much lower.

We appreciate your comment. We deleted the phrase as recommended. We further tried to reduce the complexity of our sentences throughout the manuscript.

17. Line 381: change Most to Many

Thank you for your comment. We have changed as recommended.

18. Line 405: your crowding index did not result in much risk reduction. I recommend finding out using your modeling what crowding index would make an impact. That is a much more useful result, since reducing occupancy was used a lot to reduce transmission and many studies suggested that this was an effective strategy.

Yes, thank you so much for your comment. By remarking this, you have pointed towards a mistake in the manuscript, that remained unnoticed. This mistake is mainly depicted in Figure S6, Table 2, Table 3 and further described in the last paragraph before the Discussion (“Checkerboard pattern seating”) as well as below Table 3 in the Discussion itself (“Reducing the crowding index was not a reliable mitigation measure”). We will delete the sentence in the discussion part and the latter sentence in the Results part, since it does not hold anymore. The former paragraph will be changed as follows (changes in red; line 268-273):

For MVV, halving the occupancy rate showed numerically and experimentally a similar distribution of individual infection risks P_{CFD} and P_{ATMOS} (Fig. S6, Fig. S7), but resulted in a different propagation risk pattern for MVV-2 compared to full occupancy (Fig. S8). R_{CFD} and R_{ATMOS} values resulted in a sub-proportional effect of 1.2 to 2 times for MVV-1 (Fig. S6, Fig. S7), but not exclusively for MVV-2, where the effect was 1.3 to 4.5 times (Fig. S7). While for MVV emitter position H28 and E16 the R_{CFD} values more than halved, the R_{CFD} value hardly reduced for emitter I16 (0.26 to 0.22) with the checkerboard seating arrangement (Fig. S6). In total, however, occupancy reduction is associated with a reduction in R_{CFD} for DVV and MVV.

We have also deleted the following sentence in the discussion section:

However, experimental and numerical-measurements showed a heterogeneous impact of the occupancy rate for the venues with mixing ventilation, similarly to the moderate effects of recent studies^{36,70}.

We will also update the picture in Figure S6, Table 2 and Table 3. Then, the effect of reducing the crowding index is identifiable in MVV and DVV (as also described in the text).

Personally, I would like to add a cautious note here that local effects should always be considered. Effectively, for the small risks prevalent, it is the impact of some seats nearby that drive the global risk even under mixing ventilation. Just as well, this might not be overcome in real world scenarios by reduction of occupancy.

19. Line 420, be clear that the reason that the modified analytical Wells-Riley approach did not represent a risk as well is because of the directional flow patterns resulting from the ventilation. This in turn caused spatio-temporal differences in risk.

We appreciate your comment and rephrase the sentence as recommended (changes in red; line 391-395):

“For venues with displacement or natural ventilation, however, the modified Wells-Riley approach over- or underestimated the airborne transmission risk, possibly indicating that the

strong spatio-temporal dependence of the infection risk, **caused by the directional flow patterns resulting from the ventilation**, could not be captured by this approach.”

20. One issue with the methods is that there is no assessment of uncertainty in the measurements or sensitivity to the parameters used in the models. I recommend that the authors provide an assessment of uncertainty in their predictions of risk.

Yes, we agree on this issue. Thank you for your comment. A reliable uncertainty quantification appears extremely difficult due to the multitude of influencing parameters in simulations and experiments as well as in real transmission events. For the risk assessment, the presumably most relevant parameter would be the shed viral dose of the emitter. As it is with these particular distributions, where the respiratory viral load spans many orders of magnitude (Chen et al. 2021), the goodness of prediction depends largely on the correct estimation of viral load and thereby the quanta shedding. Not least because of the attenuation of this dominant influence, we considered the effect of a high emitting scenario in our study.

In order to repair this shortcoming of missing parameter sensitivity, we will add a paragraph in the Discussion section, in which we try to outline the range of typical quanta approximations in the literature more clearly. By doing so, we hope to provide more reliable bounds for this (perhaps dominant) factor and its effect on our prediction of risk

Line 433-454:

“The effect on the resulting risk was negligible, especially, when compared to the risk differences imposed by a high emitter. However, it is crucial to note that the quanta emission rate applied in this study is itself based on certain assumptions as discussed for example in Peng et al. (2022) and Buonanno et al (2020).

Since the respiratory viral load, which is relevant for quanta emission rate, is extremely heterogeneous across individuals and spans many orders of magnitude (Chen et al. 2021), the goodness of risk prediction depends largely on the correct estimation of this influential factor. While this study used the best fit emission rate of Peng et al. (2022) (18.6 quanta h^{-1}) and standard deviation as proposed by Buonanno et al. (2020), to provide a combined lognormal distribution, other values would have been also justifiable. Aganovic et al. (2023) report the quanta emission rate to be possibly more than tenfold below that of Buonanno et al. (2020). This leads to emission rates of 0.01 quanta h^{-1} for the case of a resting and only breathing occupant at their predicted 35th percentile. On the other hand, (Li et al. 2021) estimated the median quanta emission rate in their study to be between 20 quanta h^{-1} and 454 quanta h^{-1} .

Mikszewski et al. (2022) provided values within the wide range from 0.0058 up to 4300 quanta h^{-1} depending on expiratory activity, activity level and percentile. The calculated median emission rate for a SARS-CoV-2 standing and speaking emitter was 2.7 quanta h^{-1} , while the 95th percentile was at 250 quanta h^{-1} . For a resting and oral breathing emitter the respective values were 0.55 quanta h^{-1} and 52 quanta h^{-1} . However, their listed and referenced SARS-CoV-2 outbreak events show an above-average emission rate between the 73rd and 98th percentile of the standing and speaking emitter distribution (15 quanta h^{-1} to 970 quanta h^{-1}) with one exception that is even outside the predicted range.

To conclude, the distribution assumed in this paper, although based on formerly proposed values, might be considered a rather conservative approximation with respect to a median emission scenario.”

References

Aganovic et al. (2023). New dose-response model and SARS-CoV-2 quanta emission rates for calculating the long-range airborne infection risk. *Build Environ.* 228: 109924

Buonanno et al. (2020) Quantitative assessment of the risk of airborne transmission of SARS-CoV-2 infection: prospective and retrospective applications. *Environ. Int.* 145: 106112.

Chen et al. (2021). Heterogeneity in transmissibility and shedding SARS-CoV-2 via droplets and aerosols. *Elife*. 10:e65774.

Li et al. (2021). Evaluation of infection risk for SARS-CoV-2 transmission on university campuses. *Science and Technology for the Built Environment*. 27(9): 1165-1180.

Mikszewski et al. (2022). The airborne contagiousness of respiratory viruses: A comparative analysis and implications for mitigation. *Geoscience Frontiers*. 13(6): 101285

Peng et al. (2022) Practical indicators for risk of airborne transmission in shared indoor environments and their application to covid-19 outbreaks. *Environ. Sci. Technol*. 56: 1125–1137.

21. Line 513: the term “absorbers” is not very scientific and you should use standard terminology for aerosol measurement methods. Are you using impingers to collect the aerosol? open face filters? Provide the make and manufacturer here.

Thank you for your comment. The term "absorber" was introduced in the two cited papers (Schulz & Hehnen et al. 2023; Lommel & Froese et al. 2021). The experimental measurement device used in this study was developed by Lommel & Froese et al. (2021) and is described in detail in Schulz & Hehnen et al. (2023). It is neither using impingers nor open face filters. It "absorbs" the ambient air and measures the amount of NaCl using a conductivity sensor. For the measurements NaCl serves as a tracer.

Schulz & Hehnen et al. (2021): “A vacuum pump (DC 12V 12W V, VN-C3 Mini, Vikye, China) generates a defined continuous volume flow through an inhalation tube. The volume of the aspiration airflow is 10 l min⁻¹. A fine filter (original coffee filter 1x6, Melitta, Germany) is attached to the end of the inhalation tube, surrounded by 150 ml of deionized water within a glass reservoir. The filter acts as an atomizer, breaking down the air stream into fine bubbles to dissolve the NaCl particles in the water. A peristaltic pump (DC 12 V, Vikye, China) ensures that the filter is constantly flushed, by pumping the water through a parallel circuit, in which a conductivity sensor (HI98192, Hanna Instruments, Germany) is located. The conductivity sensor has a resolution of 0.01 µS/cm and a measurement accuracy of ± 1%. The solved NaCl causes an increase of conductivity of the ultrapure water, enabling the determination of the NaCl amount.”

We have added an additional sentence for better comprehension (changes in red; line 513-517):

“Seven absorbers were distributed in the room, which inhaled the released aerosols at a flow rate of 10 l/min, generated by a vacuum pump (DC 12V 12W V, VN-C3 Mini, Vikye, China) with inhalation tube. A fine filter (original coffee filter 1x6, Melitta, Germany) is attached to the end of the inhalation tube, surrounded by deionized water and acts as an atomizer, breaking down the air stream into fine bubbles.”

References

Lommel et al. (2021) Novel measurement system for respiratory aerosols and droplets in indoor environments. *Indoor Air*. 31(6): 1860-1873

Schulz et al. (2023) Experimental device to evaluate aerosol dispersion in venues. Preprint at medRxiv. Doi: 10.1101/2023.06.13.23291335

21. Line 515: similar comment on the conductivity measurement - what instrument is this?

Thank you for your comment: The conductivity sensor is part of the aerosol measurement device called "absorber". Therefore, the product HI98192 from Hanna Instruments (Germany) was used. It has a resolution of 0.01 µS/cm and a measurement accuracy of +-1%.

We have added an additional sentence for better comprehension (line 518-519):

"Therefore, the conductivity sensor HI98192 (Hanna Instruments, Germany) with a resolution of 0.01 $\mu\text{S}/\text{cm}$ and a measurement accuracy of $\pm 1\%$ was used."

22. Line 529. Why do you have 5 significant figures for your quanta emission rate? Better to use 264 q/h

Thank you for this attentive remark! Of course, we accept your proposal and use three significant figures. However, we rounded up to 265 q/h and hope that this is acceptable for you as well

23. Line 625: "an" should be "a"

Thank you for your comment. We have changed as recommended.